# Copy number amplification of ENSA promotes the progression of triple-negative breast cancer via cholesterol biosynthesis

Yi-Yu Chen[1,2], Jing-Yu Ge[1], Si-Yuan Zhu[1], Zhi-Ming Shao[1] & Ke-Da Yu [1✉]

Copy number alterations (CNAs) are pivotal genetic events in triple-negative breast cancer (TNBC). Here, our integrated copy number and transcriptome analysis of 302 TNBC patients reveals that gene alpha-endosulfine (*ENSA*) exhibits recurrent amplification at the 1q21.3 region and is highly expressed in TNBC. ENSA promotes tumor growth and indicates poor patient survival in TNBC. Mechanistically, we identify ENSA as an essential regulator of cholesterol biosynthesis in TNBC that upregulates the expression of sterol regulatory element-binding transcription factor 2 (SREBP2), a pivotal transcription factor in cholesterol biosynthesis. We confirm that ENSA can increase the level of p-STAT3 (Tyr705) and activated STAT3 binds to the promoter of SREBP2 to promote its transcription. Furthermore, we reveal the efficacy of STAT3 inhibitor Stattic in TNBC with high ENSA expression. In conclusion, the amplification of *ENSA* at the 1q21.3 region promotes TNBC progression and indicates sensitivity to STAT3 inhibitors.

[1] Department of Breast Surgery, Shanghai Cancer Center and Cancer Institute, Shanghai Medical College, Fudan University, 200032 Shanghai, P. R. China. [2] Human Phenome Institute, Fudan University, 825 Zhangheng Road, 201203 Shanghai, P. R. China. ✉email: yukeda@fudan.edu.cn

Triple-negative breast cancer (TNBC) accounts for approximately 10–20% of all breast cancer cases. It is characterized by negative estrogen receptor (ER) and progestogen receptor (PR) expression and the lack of over-expression of HER2 (also defined by lack of ERBB2 amplification)[1]. Compared to other forms of breast cancer, TNBCs exhibit more aggressive clinical characteristics, including younger age of onset, larger tumor size, higher tumor grade, and more significant metastasis potential[2,3]. Patients with TNBC have an increased risk of distant recurrence and death within 5 years of diagnosis, and the peak of distant recurrence occurs at ~3 years[2,4]. Although chemotherapy remains the standard of care for TNBC, a subset of patients shows limited responsiveness and develops advanced diseases due to a lack of effective targeted therapy and predictive markers in this heterogeneous disease[5,6]. To solve this problem, investigators have made great efforts to elucidate the molecular nature of TNBC and seek options for molecularly targeted therapy[7–9].

Copy number alterations (CNAs) refer to somatic changes in chromosome structure, typically submicroscopic DNA alterations between 1 Kbp and 1 Mbp in length, characterized by either depletions or amplifications of DNA segments. As a hallmark of cancer, CNAs are ubiquitous in cancers and are the most common type of somatic genetic events[10,11]. Altered expression of oncogenes or tumor suppressors mediated by CNA is linked to the development, clinicopathological characteristics, and prognoses of cancers[12,13]. According to a previously defined oncogenic signature across human cancers, breast cancer falls into a class where recurrent CNAs are predominant over mutations[14]. Understanding the phenotypic effects and underlying mechanism of CNAs in breast cancer has brought advantages in targeted treatment, as highlighted by the application of trastuzumab-based therapy in HER2-overexpressing breast cancer[15]. In breast cancer, tumorigenesis driven by genomic instability is most prevalent in basal-like subtype breast cancers (most of which are TNBC), where tumors exhibit extensive CNAs[16,17]. Our previous study on a Chinese TNBC cohort revealed recurrent copy number gains in chromosomes 1q, 8q, and 10p and copy number deletions in chromosome 8p[7]. Although some reported oncogenes and tumor suppressors are affected by the most frequent CNAs, many new CNA-affected genes in TNBC remain unexplored.

Cholesterol is a precursor of bile acids, steroid hormones, vitamin D, and a component of cell membranes. It plays a critical role in cell growth and differentiation. Upregulated cholesterol biosynthesis has been discovered in several cancers, where it supports the growth, metastasis, stemness, and therapeutic resistance of tumors[18,19]. In breast cancer, cholesterol and its metabolites have been found to promote tumor progression both preclinically and clinically[20–22]. Compared to the ER + subtype, TNBC displays elevated cholesterol biosynthesis, which could have profound biological functions and indicate potential therapeutic strategies[23]. However, there is a limited understanding of how CNAs activate cholesterol biosynthesis programs in TNBC at the genetic level.

In the current study, we used integrated copy number and transcriptome analyses to discover CNA-affected oncogenes in TNBC. ENSA (alpha-endosulfine) was found to be amplified at the 1q21.3 locus in more than 18% of patients, and its amplification was correlated with increased expression in TNBC. ENSA could promote tumor growth by promoting the cholesterol biosynthesis program in TNBC. At the molecular level, the transcriptional activation of sterol regulatory element-binding transcription factor 2 (SREBP2) by phosphorylated STAT3 (p-STAT3) played a critical role in ENSA-induced cholesterol metabolism dysregulation. Consequently, hindering STAT3 phosphorylation resulted in tumor inhibition in TNBC with high

ENSA expression. These findings uncovered the mechanism by which high ENSA amplification might promote TNBC progression and suggested potential therapeutic targets.

## Results

**Integrated DNA copy number and transcriptome profiling reveals amplified ENSA at the 1q21.3 region in TNBC.** To identify dysregulated gene expression programs driven by CNAs in TNBC, we first integrated gene-level CNA data and RNA-seq data of 302 female patients from the Fudan University Shanghai Cancer Center (FUSCC) TNBC cohort to screen CNA-affected oncogenes with the criteria listed in Fig. 1a. A total of 41 genes located in several recurrent CNA regions were found (Supplementary Fig. 1). Among them, ENSA and Golgi phosphoprotein 3 like (GOLPH3L), located at the 1q21.3 peak, were the most frequently amplified genes in TNBC (amplification in 18.5% and copy number gain in 57.6% of patients), followed by genes at the 1q43 and 10p15.1 loci (Fig. 1a). In the TCGA database, the 1q21.3 segment was also frequently amplified in approximately 11% of all breast cancer patients (Fig. 1b) and predicted worse disease-free survival and disease-specific survival (Fig. 1c). However, amplification of the 1q43 and 10p15.1 regions showed no prognostic significance (Supplementary Fig. 2a). We further investigated the difference in 1q21.3 alteration between different subtypes of breast cancer in the TCGA cohort. We found a higher amplification frequency in the TNBC (27.7%) and basal-like (32.6%) subtypes (Fig. 1d), which implies that this amplicon is more critical in TNBC than in other subtypes. To identify potential tumor-promoting genes in the 1q21.3 locus of interest, we performed survival analysis of ENSA and GOLPH3L in the Kaplan–Meier plotter database and observed that elevated expression of ENSA, not GOLPH3L, was linked with poor relapse-free survival in TNBC and basal-like breast cancer (Fig. 1e, f). However, ENSA expression did not predict the survival outcomes of patients with other breast cancer subtypes, further suggesting its importance in the TNBC subtype (Supplementary Fig. 2b). ENSA expression was upregulated in several tumors and was markedly upregulated in breast cancer (Supplementary Fig. 2c). Its expression was higher in tumor tissues than in paired normal tissues and increased along with genetic amplification in TNBC (Fig. 1g, h). These results suggest that ENSA is amplified at the 1q21.3 region and is highly expressed in TNBC with clinical prognostic value.

**ENSA promotes the growth of TNBC cells.** To determine the roles of ENSA in TNBC, we knocked down its expression using shRNA and restored the expression of its most common transcript in BT549 and MDA-MB-231 cells, two TNBC cell lines with relatively high expression of ENSA (Fig. 2a and Supplementary Fig. 3a). ENSA downregulation markedly impaired the cell growth and colony formation of TNBC cells (Fig. 2b, c). However, the growth of ENSA-depleted luminal and HER2 cells was only slightly inhibited, less than the results shown in TNBC cells (Supplementary Fig. 3b). In addition, overexpression of ENSA in TNBC cells attenuated the suppression of cell growth and colony formation induced by ENSA silencing in vitro (Fig. 2d, e, and Supplementary Fig. 3c). ENSA knockdown also caused pronounced apoptosis among TNBC cells but had almost no effect on cell cycle progression (Fig. 2f and Supplementary Fig. 3d, e). These results suggest the critical role of ENSA in promoting the growth of TNBC cells.

**ENSA plays a crucial role in cholesterol biosynthesis in TNBC.** To explore the underlying molecular mechanisms of ENSA in TNBC cells, we performed RNA-sequencing analysis of both

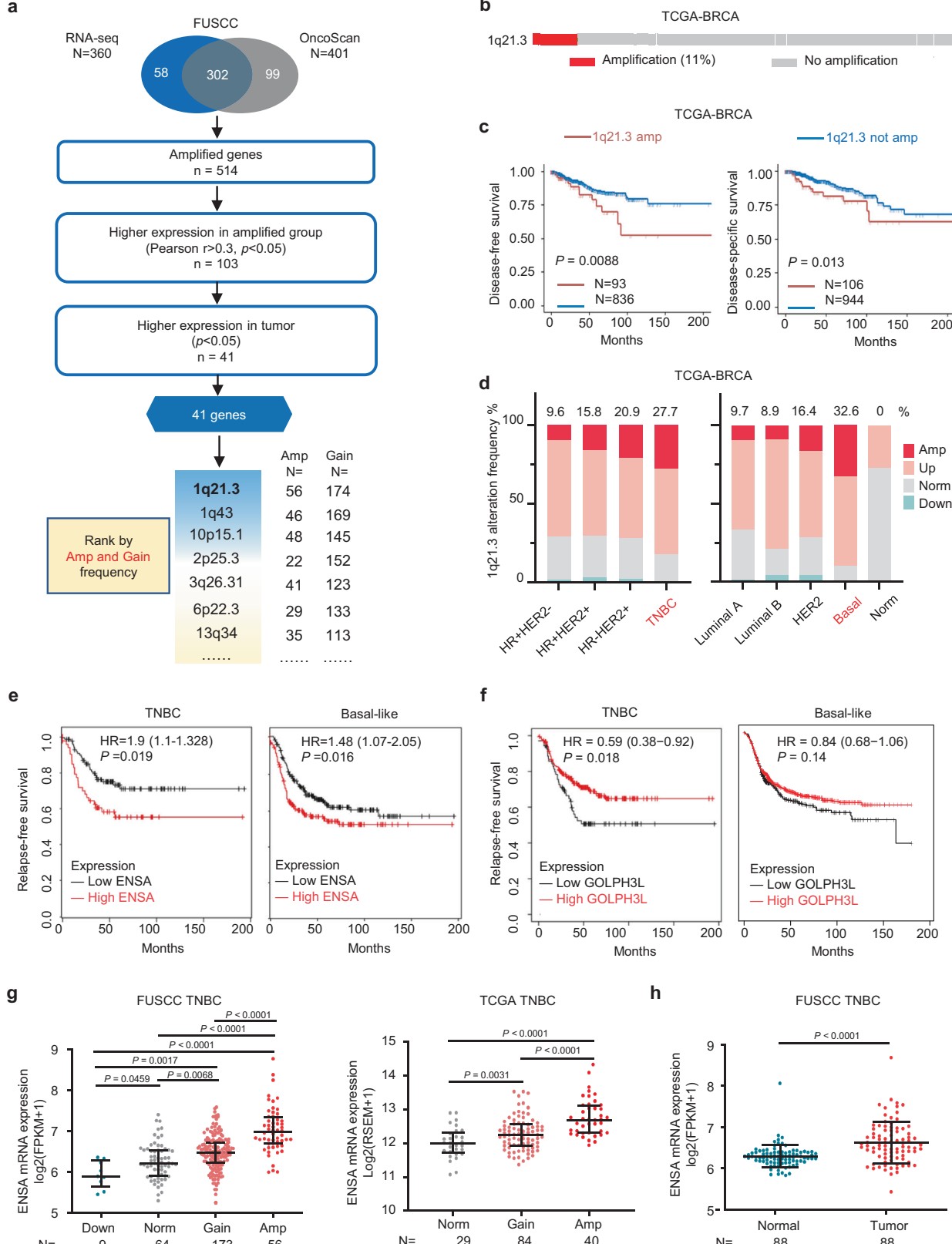

MDA-MB-231 ENSA knockdown cells and control cells. Gene set enrichment analysis (GSEA) revealed that the cholesterol pathway was the most enriched downregulated pathway in ENSA-silenced cells (Fig. 3a, b). GO analysis also indicated that the cholesterol biosynthesis pathway was strongly suppressed by ENSA knockdown (Supplementary Fig. 4a). In addition, GSEA

revealed an upregulated apoptosis pathway in ENSA-silenced cells, in accordance with the proapoptotic phenotype induced by ENSA depletion (Fig. 3c). We further explored the correlation between ENSA expression and cholesterol biosynthesis pathway activity in the FUSCC TNBC cohort using gene set variation analysis. We found a significant positive association between

**Fig. 1 ENSA is amplified at the 1.21.3 region, is highly expressed and predicts poor survival in TNBC. a** Schematic diagram depicting the screening for copy number alteration-affected genes in TNBC. **b** Copy number alteration profile of the 1q21.3 region in the TCGA breast cancer cohort. **c** Kaplan–Meier survival analysis of 1q21.3 copy number alterations in TCGA breast cancer patients. Log-rank test. **d** 1q21.3 alteration frequency in the TCGA cohort with different breast cancer subtypes. **e, f** Kaplan–Meier plots of ENSA and GOLPH3L in TNBC or basal-like BC (https://kmplot.com/analysis/). Log-rank test. **g** ENSA expression of samples with different ENSA copy number in FUSCC and TCGA TNBC cohorts. n = 302 in FUSCC TNBC cohort and n = 153 in TCGA TNBC cohort. Data are presented as mean ± SD. Two-tailed one-way ANOVA tests and adjustments were made for multiple comparisons. **h** ENSA expression of 88 paired tumor tissues versus adjacent normal tissues in FUSCC cohort. n = 88 paired samples. Data are presented as mean ± SD. Two-tailed paired Student's t test. Source data are provided as a Source Data file. FUSCC Fudan University Shanghai Cancer Center, TCGA The Cancer Genome Atlas, TNBC triple-negative breast cancer, BRCA breast cancer, FPKM fragments per kilobase million, RSEM RNA-seq by expectation maximization, Amp amplification, Norm normal.

ENSA mRNA expression and the cholesterol biosynthesis program in TNBC (Fig. 3d), which was further validated in The Cancer Genome Atlas (TCGA), Molecular Taxonomy of Breast Cancer International Consortium (METABRIC) and Korean breast cancer (SMC) cohorts (Supplementary Fig. 4b). Decreased expression of cholesterol biosynthesis enzymes and SREBP2, a crucial transcription factor that preferentially activates genes involved in cholesterol biosynthesis, was further validated at the mRNA and protein levels in TNBC cells expressing ENSA shRNA (Fig. 3e, f). Importantly, the protein levels of the full and cleaved forms of SREBP2 were both decreased, which was consistent with the decreased mRNA level of SREBP2 upon ENSA silencing (Fig. 3f). Overexpression of ENSA in ENSA knockdown TNBC cells restored the expression of cholesterol biosynthesis genes (Supplementary Fig. 4c). Intriguingly, the expression of SREBP2 and downstream enzymes was not altered by ENSA knockdown in non-TNBC cell lines, suggesting a critical role of the ENSA-regulated cholesterol pathway in TNBC (Supplementary Fig. 4d). To determine whether the main products of the cholesterol biosynthesis pathway had changed, we detected the contents of cholesterol and intermediate metabolites by liquid chromatography-mass spectrometry. Consistent with the down-regulated mRNA and protein expression of genes involved in cholesterol biosynthesis, the concentration of total cholesterol and most intermediates decreased upon ENSA silencing in TNBC cells (Fig. 3g, h). In addition, ENSA knockdown caused a significant decrease in free cholesterol content in TNBC cells (Fig. 3i). We next examined whether the ENSA knockdown-induced phenotype was attributable to cholesterol depletion. The addition of cholesterol (2.5 μg/ml) partially attenuated the inhibition of cell growth and increased apoptosis in ENSA-depleted TNBC cells (Fig. 3j and Supplementary Fig. 4e, f). Together, these findings support the notion that ENSA might promote the growth of TNBC cells by regulating cellular cholesterol biosynthesis.

**Activated STAT3 is involved in ENSA-induced tumor growth.** To elucidate the molecular mechanism of the ENSA depletion-induced phenotype in TNBC cells, we performed transcription factor (TF) analysis of the RNA-seq data and hypothesized that STAT3 was a key TF responsible for the downstream alteration (Fig. 4a). In TNBC cells, p-STAT3 (Tyr705), but not p-STAT3 (Ser727) or total STAT3, was markedly decreased upon ENSA knockdown (Fig. 4b). Overexpression of ENSA restored the level of p-STAT3 (Tyr705) (Fig. 4c). Next, we confirmed the involvement of STAT3 in ENSA knockdown-induced growth inhibition by overexpressing STAT3 in ENSA-silenced TNBC cells to induce the constitutive activation/phosphorylation of STAT3. Our results showed that the overexpression of STAT3 could partially attenuate TNBC cell growth inhibition induced by ENSA depletion (Fig. 4d, e).

In mouse mammary fat pad xenografts, ENSA knockdown resulted in a significant reduction in the tumor growth of MDA-MB-231 cells, which could be completely rescued by

overexpression of ENSA and mostly rescued by overexpression of STAT3 (Fig. 4f, g, and Supplementary Fig. 5a). Immunohistochemical (IHC) staining of p-STAT3 (Tyr705) significantly decreased upon ENSA silencing. However, in ENSA-depleted TNBC tumors, overexpression of either ENSA or STAT3 increased p-STAT3 (Tyr705) staining (Fig. 4h and Supplementary Fig. 5b). In accordance with the proapoptotic phenotype induced by ENSA depletion in vitro, IHC staining of cleaved caspase 3 displayed a corresponding change in each group of xenografts (Fig. 4h and Supplementary Fig. 5c). In addition, we also observed decreased lung metastasis incidence in the ENSA-depleted group (Supplementary Fig. 5d). Taken together, these data suggest the critical role of ENSA-STAT3 signaling in promoting the growth of triple-negative tumors.

**ENSA activates STAT3 to regulate cholesterol biosynthesis.** We next questioned whether activated STAT3 participated in ENSA-induced alterations in the cholesterol biosynthesis pathway. The prediction from the JASPAR database showed that STAT3 was likely to bind to the promoter of SREBP2 (Fig. 5a). ChIP-seq data analysis from GSE152203 also showed a STAT3-binding peak in SREBP2 in MDA-MB-231 cells (Supplementary Fig. 6a). Using a ChIP assay, we confirmed that STAT3 occupied a site in the promoter region of SREBP2 (Fig. 5b). Moreover, STAT3 silencing in TNBC cells suppressed the mRNA and protein expression of SREBP2 (Fig. 5c and Supplementary Fig. 6b). These results indicate that STAT3 could bind to the promoter of SREBP2 and alter the expression of SREBP2 at the transcriptional level in TNBC cells. We next validated that the promoter activity of SREBP2 was restrained by ENSA depletion and rescued by STAT3 overexpression based on the luciferase reporter assay (Fig. 5d). Consistently, the expression level of SREBP2 and enzymes involved in cholesterol biosynthesis decreased after ENSA knockdown, but this effect was reversed by the overexpression of STAT3 (Fig. 5e). In addition, the cellular free cholesterol levels, which were decreased with ENSA depletion, could also be restored by overexpression of STAT3 (Fig. 5f and Supplementary Fig. 6c). In mouse mammary fat pad xenograft models, the expression of SREBP2 and FDPS decreased after ENSA knockdown and was restored when exogenous STAT3 was overexpressed (Supplementary Fig. 6d).

We next explored whether the inhibitory effect of ENSA knockdown on p-STAT3 was dependent on protein phosphatase 2A (PP2A), a known protein phosphatase whose function is inhibited by direct interaction with ENSA. We silenced PP2A using its catalytic subunit-targeting siRNA (siPPP2CA) in MDA-MB-231 cells and observed an increase in p-STAT3 (Tyr705) expression (Fig. 5g). However, p-STAT3 (Ser727) was not influenced by PP2A silencing. Furthermore, the inhibition of p-STAT3 (Tyr705) and SREBP2 expression by ENSA depletion was weakened by an additional knockdown of PP2A in TNBC cells (Fig. 5h). These results suggest that ENSA might promote

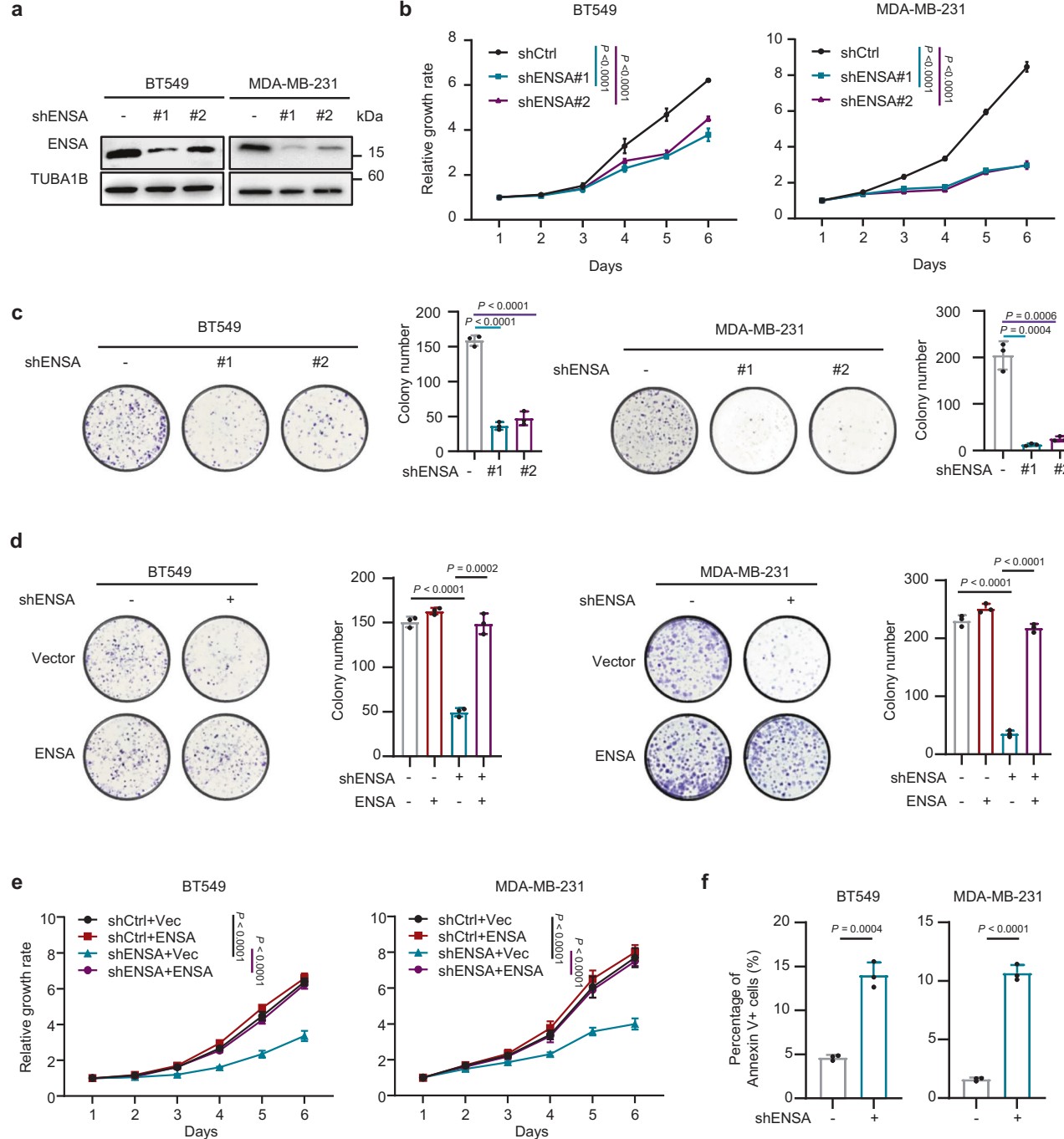

**Fig. 2 ENSA is a major driver of TNBC cell growth. a** Stable silencing of ENSA expression in the TNBC cell lines BT549 and MDA-MB-231. **b** In vitro growth curves of BT549 and MDA-MB-231 cells expressing control or ENSA shRNA. $n = 6$. Data are presented as mean ± SD. Two-tailed two-way ANOVA tests. **c** Colony formation of BT549 and MDA-MB-231 cells expressing control or ENSA shRNA. $n = 3$. Data are presented as mean ± SD. Two-tailed unpaired Student's $t$ tests. **d** Colony formation of BT549 and MDA-MB-231 cells ± ENSA knockdown and ± ENSA overexpression. $n = 3$. Data are presented as mean ± SD. Two-tailed unpaired Student's $t$ tests. **e** In vitro growth curves of BT549 and MDA-MB-231 cells ± ENSA knockdown and ± ENSA overexpression. $n = 6$. Data are presented as mean ± SD. Two-tailed two-way ANOVA tests. **f** Apoptosis levels were measured in BT549 and MDA-MB-231 cells expressing control or ENSA shRNA. Percentage of annexin V+ cells are shown. $n = 3$. Data are presented as mean ± SD. Two-tailed unpaired Student's $t$ tests. Source data are provided as a Source Data file.

the phosphorylation of STAT3 to regulate cholesterol biosynthesis in TNBC cells in a PP2A-dependent manner.

**ENSA determines therapeutic sensitivity to STAT3 inhibitors.** As ENSA regulates the activity of STAT3, we sought to evaluate whether ENSA could be a therapeutic marker of STAT3 inhibitor sensitivity in TNBC cells. We used Stattic as a small molecule

inhibitor of STAT3 activation. Compared to the ENSA knockdown group, the control group with higher ENSA levels was more sensitive to Stattic and exhibited significantly reduced growth after Stattic treatment (Fig. 6a, b, and Supplementary Fig. 7a, b). In organoids derived from three TNBC patients with diverse levels of ENSA expression, we also found that the sensitivity of organoids to Stattic increased along with the increase in the

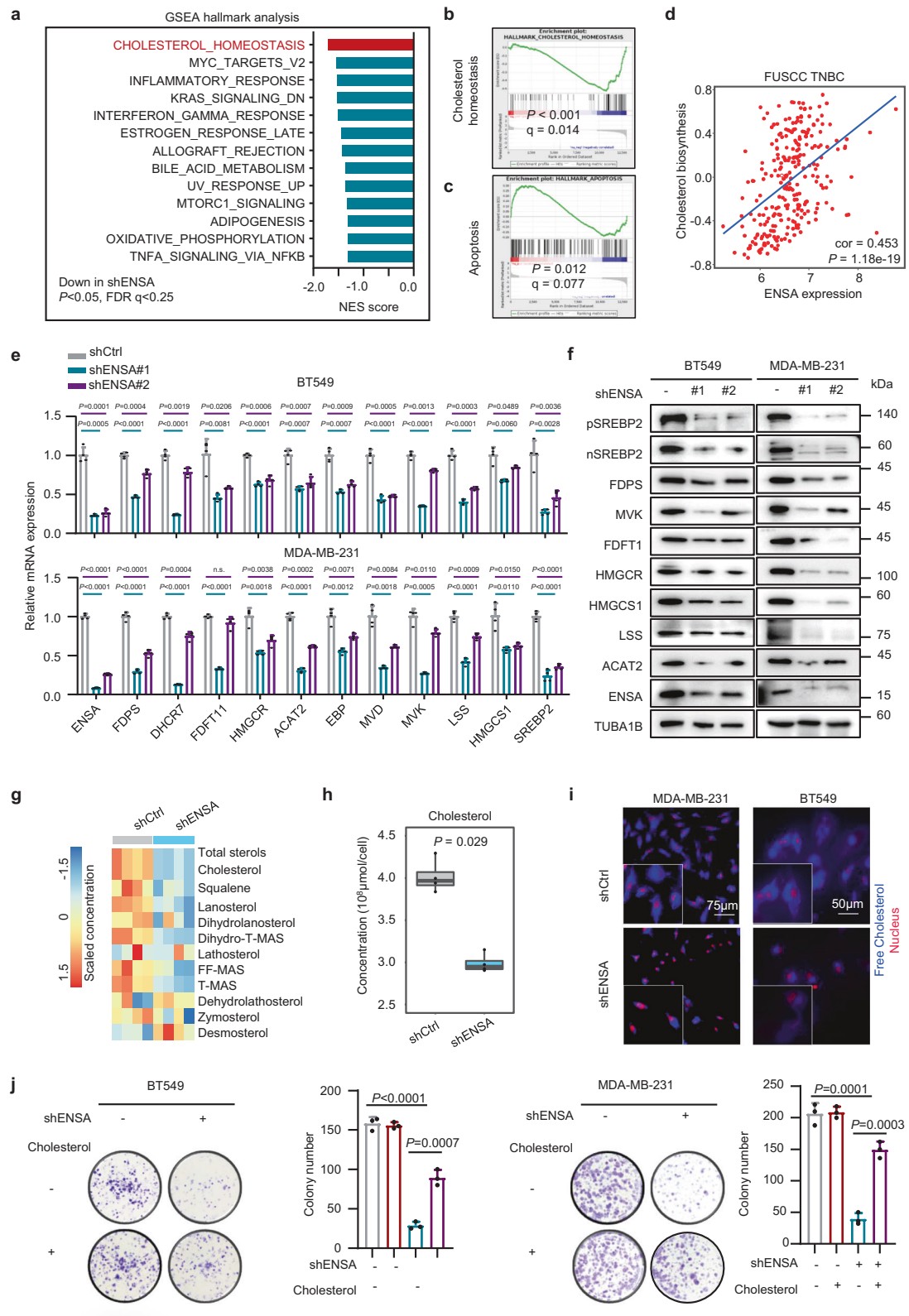

ENSA expression level (Fig. 6c–e and Supplementary Fig. 7c). The clinicopathological features of these three organoids are shown in Supplementary Table 1. We next explored the sensitivity of TNBC cells to Stattic in xenograft models and found significantly impaired sensitivity to Stattic when ENSA was depleted (Fig. 6f, g, and Supplementary Fig. 7d, e). To further explore the drug response in patients with TNBC, we constructed seven mini-patient-derived xenograft (mini-PDX) models, as reported previously[24,25], and measured the response for Stattic normalized to vehicle treatment (Fig. 6h). The clinicopathological features of these TNBC models are shown in Supplementary Table 2. In line with the results obtained above, ENSA-high tumors (578, 573,

**Fig. 3 ENSA plays a crucial role in cholesterol biosynthesis in TNBC. a** GSEA of downregulated pathways after ENSA knockdown in MDA-MB-231 cells. The top 10 pathways ($P < 0.05$ and FDR q < 0.25) ranked by absolute normalized enrichment scores are shown. NES score and nominal *P*-value were given by GSEA software. **b** Enrichment plot of the cholesterol homeostasis pathway after ENSA knockdown in MDA-MB-231 cells. **c** Enrichment plot of the apoptosis pathway after ENSA knockdown in MDA-MB-231 cells. **d** Scatter plot showing the correlation of ENSA expression with the cholesterol biosynthesis pathway score in FUSCC TNBC data identified by 'gsva' method. Correlation coefficients were calculated using the *Pearson* test. Two-tailed *P*-values were given. **e** qRT-PCR analysis of the relative transcript levels of cholesterol biosynthesis pathway genes after ENSA knockdown in MDA-MB-231 cells. $n = 4$. Data are presented as mean ± SD. Two-tailed unpaired Student's *t* tests. **f** Western blotting images showing proteins involved in the cholesterol biosynthesis pathway after ENSA knockdown in MDA-MB-231 cells. $n = 3$ independent experiments. **g** Heatmap displaying the concentration of cholesterol and intermediates in the cholesterol biosynthesis pathway after ENSA knockdown in MDA-MB-231 cells. $n = 4$. **h** Total cellular cholesterol contents in MDA-MB-231 cells were analyzed by LC-MS with normalization to cell quantity. $n = 4$. The center line corresponded to the median, the lower and upper hinges corresponded to the first and third quartiles, and the upper/lower whisker extends from the hinge to the largest/smallest value no further than 1.5 times interquartile range. Two-tailed unpaired Wilcoxon test. **i** Filipin III staining showing the cellular free cholesterol content in MDA-MB-231 and BT549 cells with ENSA knockdown. $n = 3$ independent experiments. **j** Colony formation of BT549 and MDA-MB-231 cells expressing control or ENSA shRNA after treatment with 2.5 μg/ml exogenous cholesterol. $n = 3$. Data are presented as mean ± SD. Two-tailed unpaired Student's *t* tests. Source data are provided as a Source Data file. GSEA gene set enrichment analysis, NES normalized enrichment score.

---

543, 553) showed higher sensitivity to Stattic than those with relatively low-ENSA expression (554, 584, 552) (Fig. 6i, j, and Supplementary Fig. 7f, g). Together, the results from our TNBC cell lines, organoids, animal models and mini-PDX models strongly suggest that ENSA expression can be a biomarker for effective treatment with STAT3 inhibitors.

**Correlations among ENSA, p-STAT3, and SREBP2 expression in clinical samples and patient outcomes.** To investigate the clinical relevance of our findings, we first evaluated the protein expression levels of ENSA in 8 paired primary TNBC specimens and adjacent normal tissues by IHC assay. The results revealed that the ENSA protein levels were much higher in TNBC specimens than in normal tissues (Fig. 7a, b). To investigate the correlation of ENSA protein expression with patient survival, we collected surgical samples from 138 TNBC patients and detected the expression level of ENSA by performing IHC analysis. Representative IHC images are shown in Fig. 7c. Kaplan–Meier analysis of specimens revealed that patients harboring tumors with high ENSA levels tended to have worse relapse-free survival and overall survival than patients harboring low-ENSA levels (Fig. 7d, $P < 0.05$). Multivariable analysis showed that ENSA expression still showed prognostic value for relapse-free survival ($P = 0.04$) and tended to correlate with worse overall survival ($P = 0.07$) after adjustment for age, tumor size, and lymph node status (Supplementary Tables 3 and 4). We further examined the protein expression levels of SREBP2, HMGCR, and p-STAT3 (Tyr705). Representative IHC images are shown in Fig. 7e and Supplementary Fig. 8a. The IHC results showed that ENSA expression was positively correlated with p-STAT3 (Tyr705), SREBP2, and HMGCR (Fig. 7f and Supplementary Fig. 8b). In addition, the survival analysis of SREBP2 expression showed that high SREBP2 expression was correlated with worse relapse-free survival of TNBC patients in internal and external cohorts (Supplementary Fig. 8c, d). The expression correlation between ENSA and SREBP2 was also validated in an external cohort (Supplementary Fig. 8e). Together, these results indicate that the expression of ENSA is positively correlated with that of downstream molecules and is a poor prognostic factor in TNBC.

## Discussion
Multiple lines of evidence have proven that CNAs promote the initiation and progression of cancers by altering the expression levels of oncogenes and tumor suppressors. In this study, we found significant amplification of the chromosome 1q21.3 region in TNBC, and gene *ENSA* at this locus was highly expressed. ENSA regulates TNBC cell growth in vitro and in vivo through

STAT3-mediated transcriptional activation of SREBP2 and downstream cholesterol biosynthesis (Fig. 7g).

TNBC exhibits high genomic instability, resulting in frequent CNAs at the chromosome level. Identification of the CNA-driven phenotype and the underlying mechanisms provides new insights into the pathogenesis of TNBC and facilitates the discovery of therapeutic treatments. The gain of 1q is one of the most frequent genomic imbalances in breast carcinomas and exhibits a higher CNA frequency in basal tumors than in luminal tumors[26–28]. Our previous research on the FUSCC cohort also indicated the widespread occurrence of 1q amplification in Chinese TNBC patients[7]. Here, we further showed that 1q21.3 was the exact region that harbored the highest frequency of gains in the 1q chromosome band (>18% in FUSCC TNBC patients), and this alteration frequency was just below that of the 8q amplicon (where *MYC* is located). Based on the analysis of the TCGA breast cancer cohort, we found amplification of the 1q21.3 region in all breast cancer subtypes, but the amplification rate was higher in TNBC than in other subtypes, which suggests the significance of this amplicon in TNBC. Evidence has indicated that CNAs promote the progression of tumors by altering the expression of genes within those affected genomic regions[29]. For example, 1q21.3 amplification upregulates the expression of several encompassing genes that form a regulatory loop to drive tumor growth[30]. Herein, we utilized copy number and gene expression data of primary tumors, including 88 paired tumor tissues and peritumor tissues from the FUSCC TNBC cohort, to discover potential oncogenes within amplicons. Among the 41 candidates identified, *ENSA* was located in the frequently amplified 1q21.3 region and indicated a poor prognosis in TNBC.

The *ENSA*-encoding protein belongs to the highly conserved c-AMP-regulated phosphoprotein (ARPP) family and was initially identified as an endogenous ligand for the sulfonylurea receptor that modulates insulin secretion and glucose metabolism[31,32]. It has previously been shown that in Xenopus egg extracts, ENSA and its close relative ARPP19 are substrates of great wall (GWL) kinase and act as competitor inhibitors to prevent PP2A-B55 from dephosphorylating substrates such as cyclin B-CDK1, which results in mitotic entry[33,34]. A subsequent study on human cells identified the ability of ENSA to control the length of the S phase[35]. While GWL and PP2A have been heavily studied, the genomic alteration and molecular function of ENSA have rarely been reported in the context of cancer[36]. To test these findings, we generated ENSA knockdown TNBC cells and found that these cells exhibited pronounced growth inhibition. Surprisingly, altered expression of ENSA did not significantly influence the cell cycle distribution of TNBC cells at the S phase or G2/M phase, which implies that the function of ENSA is context-dependent. Interestingly, cholesterol

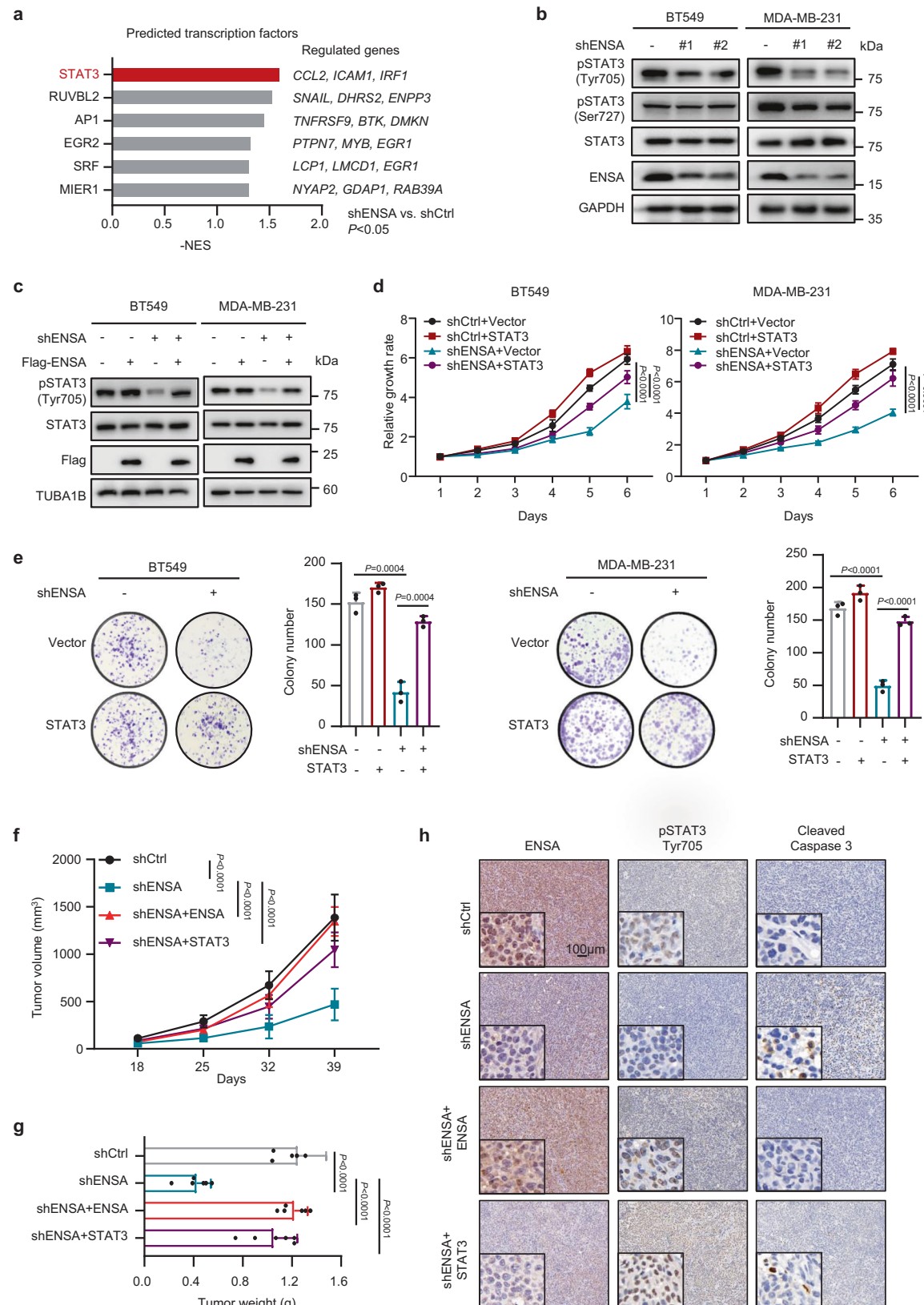

biosynthesis was the most enriched pathway, and cholesterol addition abolished the impaired growth effect induced by ENSA knockdown in TNBC cells. These preliminary results link ENSA to lipid metabolic programming in TNBC.

The mevalonate (MVA) pathway is a critical metabolic pathway responsible for de novo cholesterol biosynthesis and the

production of other metabolites, such as isoprenoids, dolichol, and ubiquinone, to support multiple cellular processes[37]. The importance of the MVA pathway and its metabolites in supporting cancer cell survival and growth has been increasingly appreciated[38]. Cholesterol is a vital metabolite for the biological functions of mammalian cells. Its concentration at both the

**Fig. 4 ENSA promotes tumor growth by activating STAT3. a** Candidate transcription factor (TF) prediction performed by GSEA of regulatory target gene sets. The top 6 TFs ($P < 0.05$) ranked by absolute normalized enrichment scores are shown. NES score and nominal $P$-value were given by GSEA software. **b** Western blotting images showing the protein levels of phosphorylated STAT3 (pSTAT3)-Tyr705, pSTAT3-Ser727 and total STAT3 after ENSA knockdown in BT549 and MDA-MB-231 cells. $n = 3$ independent experiments. **c** Western blotting images showing the protein levels of pSTAT3-Tyr705 and total STAT3 in BT549 and MDA-MB-231 cells ± ENSA knockdown and ± ENSA overexpression. $n = 3$ independent experiments. **d** In vitro growth curves of BT549 and MDA-MB-231 cells ± ENSA knockdown and ± STAT3 overexpression. $n = 6$. Data are presented as mean ± SD. Two-tailed two-way ANOVA tests. **e** Colony formation of BT549 and MDA-MB-231 cells ± ENSA knockdown and ± STAT3 overexpression. $n = 3$. Data are presented as mean ± SD. Two-tailed unpaired Student's $t$ tests. **f** In vivo growth curve of tumors ($n = 6$) generated by injecting MDA-MB-231 cells expressing control or ENSA shRNA and rescued by ENSA or STAT3 overexpression. $n = 6$ mice per group. Data are presented as mean ± SD. Two-tailed two-way ANOVA tests. **g** Tumor weight of MDA-MB-231 cells ($n = 6$) expressing control or ENSA shRNA rescued by ENSA or STAT3 overexpression. $n = 6$ mice per group. Data are presented as mean ± SD. Two-tailed unpaired Student's $t$ tests. **h** Immunohistochemical images of ENSA, pSTAT3-Tyr705, and cleaved caspase 3 in mammary fat pad xenograft models. Scale bar: 100 μm. Source data are provided as a Source Data file. NES normalized enrichment score.

cellular and systemic levels has been linked to many diseases, such as obesity, heart disease and cancer. Several groups have addressed the relationship between hypercholesterolemia and increased breast cancer risk based on clinical data[21,39,40]. Importantly, 27-hydroxycholesterol is a key molecule that links hypercholesterolemia with breast cancer pathophysiology[20]. In addition to systemic cholesterol, dysregulated cellular cholesterol, derived from increased biosynthesis or uptake, fuels the malignant phenotypes of cancer cells, including proliferation, anti-apoptosis, migration, stemness, and immune escape[41–46]. Several other MVA pathway metabolites and enzymes have also been identified as oncogenic. For example, MVA-derived farnesyl-diphosphate and geranylgeranyl-diphosphate are critical for the isoprenylation of proteins, supporting their proper localization and function in cancer cells[38]. Metabolite quinone coenzyme Q is involved in cancer cell energy metabolism[47]. High-resolution CRISPR screens also identified several MVA pathway enzymes essential for the survival of cancer cells[48]. In the current study, we found significantly decreased concentrations of cholesterol and suppressed expression of cholesterol biosynthesis enzymes upon ENSA silencing. However, the addition of cholesterol alone was not able to completely rescue impaired cell growth induced by ENSA depletion, implying the possibility of other MVK pathway enzymes or metabolites regulating ENSA-induced cell growth. Together, these results suggest the importance of the MVK pathway, primarily cholesterol biosynthesis, in the regulation of TNBC growth by ENSA.

De novo cholesterol biosynthesis is mainly controlled by SREBP2, the pivotal transcription factor for genes encoding enzymes involved in the cholesterol biosynthesis program[49]. SREBP-2 is synthesized as a 125 kDa inactive precursor and sequentially cleaved into the NH2-terminal form with nuclear translocation and transcription factor activity[50]. It was previously found that SREBP2 was altered by several oncogenic pathways, including p53, PI3K/AKT/mTOR, and AMPK[51–55]. However, the relationship between STAT3 and cholesterol biosynthesis remains unclear. Only a few studies have described a decrease in the mRNA expression of SREBP1/2 with the deletion of STAT3, but there is limited knowledge on the underlying mechanism[56,57]. Intriguingly, we used ChIP-qPCR to identify that STAT3 bound directly to the promoter sequences of SREBP2 and promoted the transcription of SREBP2 in TNBC cells. The decrease in tumor growth and the expression of SREBP2 by ENSA silencing could be abolished by ectopic STAT3 expression, which confirmed the central role of STAT3 in cholesterol biosynthesis in TNBC. Additionally, the STAT3-independent pathway cannot be neglected since ectopic STAT3 expression alone could rescue most of but not all the growth inhibition induced by ENSA depletion. Other important oncogenic pathways, such as the MYC pathway, were also enriched in ENSA-depleted TNBC cells. Taken together, ENSA might promote TNBC progression

through multiple pathways, of which the STAT3-SREBP2 axis was the most important.

The phosphorylation level of STAT3 can be regulated in different ways. On the one hand, phosphorylation of STAT3 at Tyr705 by tyrosine kinases such as JAK and SRC or at Ser727 by JNK and other MAPKs results in its activation in cancer. On the other hand, inhibition of negative regulators such as PIAS3, SOCS1 and 3 and several cellular phosphatases (SHP1 and 2, PTPRD, PTPRT, PTPN1 and 2, DUSP22) can also lead to STAT3 activation in cancer. To determine how ENSA impacts the phosphorylation of STAT3, we focused on PP2A, a serine/threonine phosphatase whose functions can be suppressed by direct interaction with ENSA. It has been reported that pharmacologic inhibition of PP2A induces the phosphorylation of STAT3 on serine residues in T cells and vascular smooth muscle cells[58,59]. Unlike previous studies, we found that genetic inhibition of the PP2A catalytic subunit in control or ENSA knockdown TNBC cells could alter the phosphorylation of STAT3 at the Tyr705 residue instead of the Ser727 residue, which implied that ENSA-PP2A affects p-STAT3 (Tyr705) expression in TNBC cells. The reason the observations for PP2A were different in our study might be related to the distinct cellular context, and PP2A might indirectly impact the tyrosine residue of STAT3 in TNBC cells. The underlying mechanism by which ENSA-PP2A acts on the tyrosine residue of STAT3 remains to be addressed in the near future.

Some limitations of the current study should be acknowledged. Firstly, one cell line derived xenograft models and a limited number of mini-PDX models might not fully represent tumor features in vivo, more preclinical models are needed to verify the conclusions of this study. Secondly, while mini-PDX models could overcome the time-consuming disadvantage of traditional PDX models and retained the accuracy and efficiency in drug sensitivity testing, they still have some limitations compared to PDX models. Unlike PDX models which represent a more realistic tumor microenvironment, tumors grown as mini-PDXs lack interactions with human microenvironmental components due to the removal of blood cells and fibroblasts in the sample preparation process[60]. Thus, the involvement of microenvironmental components in ENSA-induced tumor progression and drug sensitivity deserve further investigation in proper preclinical models.

In summary, our current research reveals that *ENSA*, a gene with recurrent CNA at the 1q 21.3 locus, is a trigger for tumor growth that acts by promoting cholesterol biosynthesis in TNBC. Further characterization of the potential mechanism of the ENSA-PP2A-STAT3-SREBP2 regulatory axis might support our findings. We propose that the STAT3 inhibitor Stattic might be an apt option for treating ENSA-expressing TNBC.

## Methods

**Clinical samples.** FUSCC TNBC cohort data (Sequence Read Archive (SRA) dataset: SRP157974; Gene Expression Omnibus (GEO) dataset: GSE118527) and

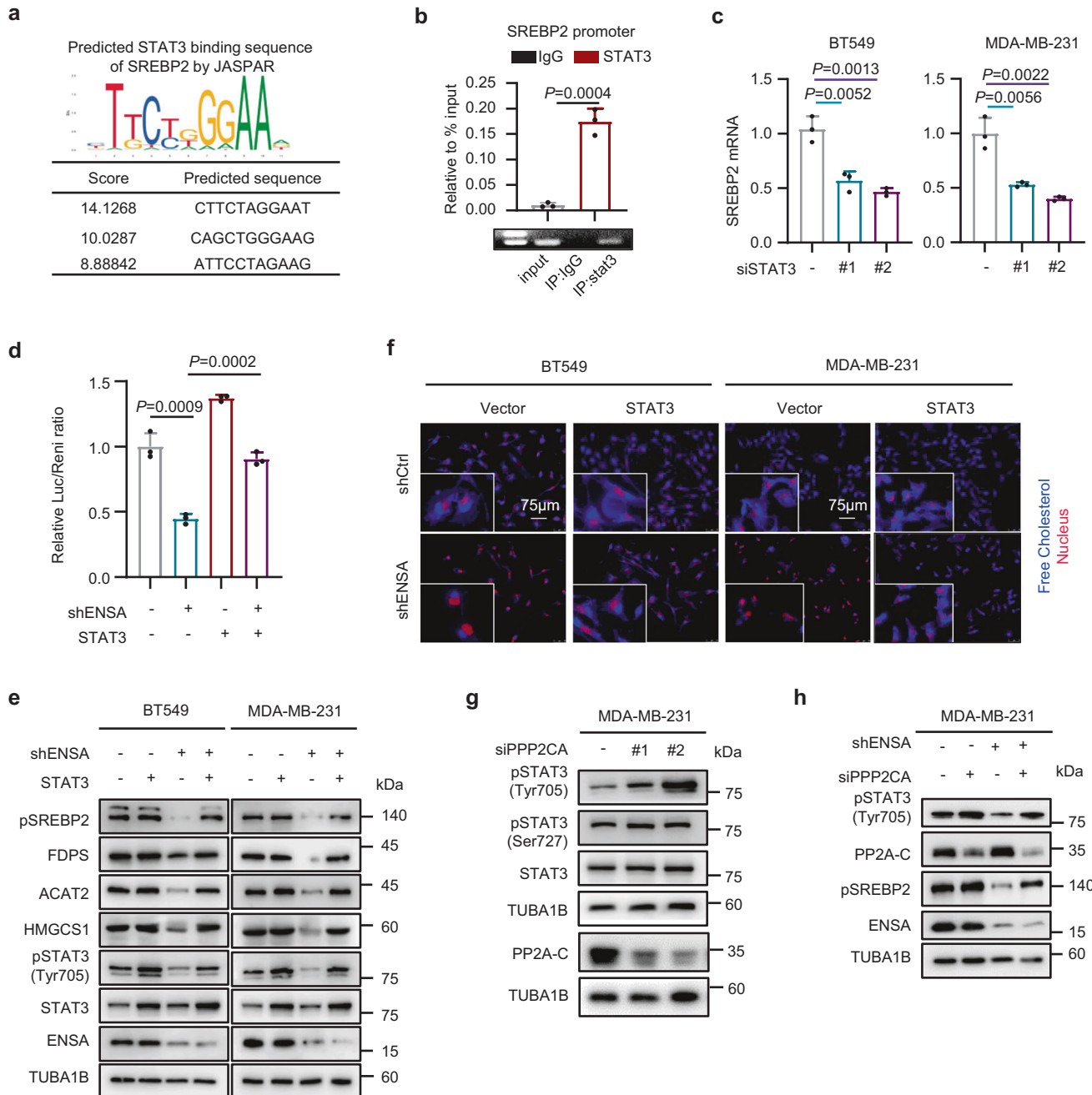

**Fig. 5 ENSA activates STAT3 to promote the transcription of SREBP2 in a PP2A-dependent manner. a** JASPAR prediction of STAT3-binding sites on the sequence of SREBP2. **b** qRT-PCR and PCR analysis of STAT3 at the SREBP2 promoter after ChIP assays in MDA-MB-231 cells expressing control or ENSA shRNA. $n = 3$. Data are presented as mean ± SD. Two-tailed unpaired Student's $t$ tests. **c** qRT-PCR detecting relative SREBP2 mRNA expression in BT549 and MDA-MB-231 cells after STAT3 transient silencing. $n = 3$. Data are presented as mean ± SD. Two-tailed unpaired Student's $t$ tests. **d** Luciferase reporter assay detecting the activity of the SREBP2 promoter in BT549 cells ± ENSA knockdown and ± STAT3 overexpression. $n = 3$. Data are presented as mean ± SD. Two-tailed unpaired Student's $t$ tests. **e** Western blotting images showing the expression of enzymes involved in cholesterol biosynthesis, SREBP2, pSTAT3-Tyr705 and total STAT3 in BT549 and MDA-MB-231 cells with ± ENSA knockdown and ± STAT3 overexpression. $n = 3$ independent experiments. **f** Filipin III staining showing the cellular free cholesterol contents of BT549 and MDA-MB-231 cells ± ENSA knockdown and ± STAT3 overexpression. $n = 3$ independent experiments. **g** Western blotting images showing the expression of STAT3, pSTAT3-Tyr705, and pSTAT3-Ser727 in MDA-MB-231 cells expressing control or PPP2CA siRNA. $n = 3$ independent experiments. **h** Western blotting images showing the expression of pSTAT3-Tyr705 and SREBP2 in MDA-MB-231 cells ± ENSA knockdown and ± transient PPP2CA knockdown. $n = 3$ independent experiments. Source data are provided as a Source Data file.

data analyses were performed according to a previous study[7]. In brief, 465 female Chinese TNBC patients who underwent surgery at FUSCC were retrospectively selected, and their RNA-sequencing (RNA-seq) data, whole-exome sequencing, and OncoScan microarray copy number data were obtained. Among these 465 patients, 302 patients who had both RNA-seq data and copy number data were included in our study for screening candidate CNA-affected genes. Analysis of gene-level CNAs was performed according to a previous study[7]. In brief, probe-level output from the OncoScan Console was analyzed by ASCAT (v2.4.3), and the produced segment data were imported into GISTIC2.0 (v2.0.22) to acquire gene-level CNA data.

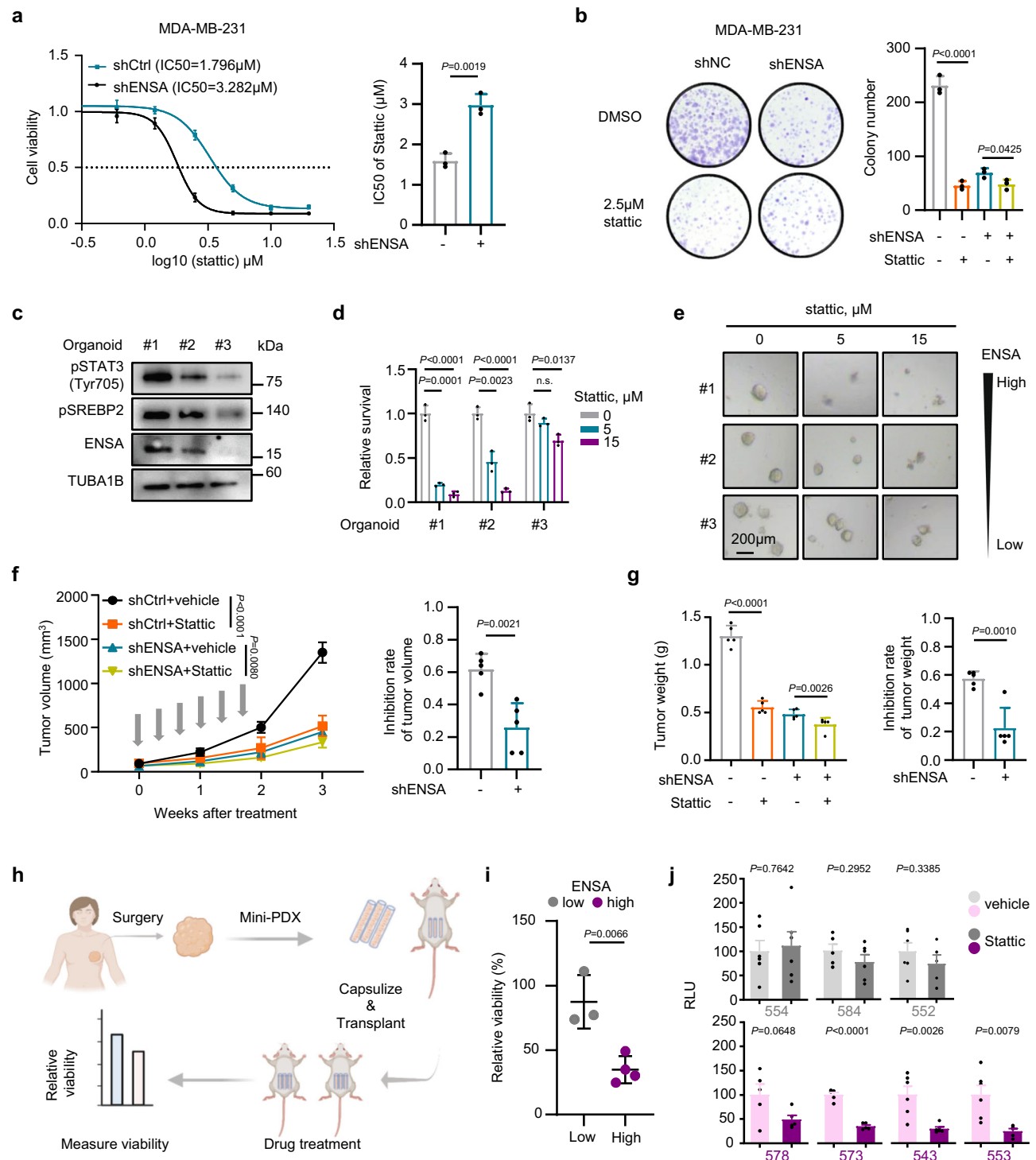

**Bioinformatic analyses**. Gene set variation analysis of FUSCC TNBC transcriptome data was carried out to calculate the enrichment score of the cholesterol biosynthesis pathway in each sample with the R (v4.0.3) 'GSVA' package. The Pearson correlation metric was computed to assess the associations among the expression of different genes and between gene expression levels and cholesterol synthesis pathway scores by using the 'cor' function in R.

**Cell lines**. The TNBC cell lines BT549, MDA-MB-231, MCF7, T47D, SKBR3, and BT474 and embryonic kidney cells (HEK293T) were obtained from the American Type Culture Collection (ATCC) and cultured in DMEM supplemented with 10% FBS. The cell lines were regularly confirmed to be negative for mycoplasma contamination with a Mycoplasma Detecting Kit (Vazyme).

**Short hairpin RNA (shRNA) vectors and lentiviral infections of cells**. The two shRNAs with the best knockdown efficiency (shENSA-1: GAGCTGAAGAGG CAAAGCTAA; shENSA-2: CTGCCAGATCCTGAGACGCTT) were cloned into the pLKO.1 vector and introduced into HEK293T cells together with packing plasmids (psPAX2 and pMD2. G) and standard Lipofectamine 2000 transfection reagent (Thermo Fisher Scientific) to generate lentiviruses. The viral supernatants were collected and applied to infect TNBC cell lines in the presence of polybrene (10 µg/ml; Sigma-Aldrich). A plasmid expressing a nontargeting shRNA was used as the negative control.

**Small interfering RNA (siRNA) delivery**. siRNAs targeting human STAT3 (siSTAT3-1: GCAAAGAATCACATGCCACTT; siSTAT3-2: GGCGTCCAGT

**Fig. 6 ENSA is linked to Stattic sensitivity in TNBC. a** Dose–response curves and half maximal inhibition concentration values of Stattic in MDA-MB-231 cells expressing control or ENSA shRNA. Dose-response curves: $n = 6$; Data are presented as mean ± SD. Bar plot: $n = 3$ independent experiments; Data are presented as mean ± SD; Two-tailed unpaired Student's $t$ test. **b** Clonogenic survival assays of MDA-MB-231 cells expressing control or ENSA shRNA and treated with 2.5 µM Stattic. $n = 3$. Data are presented as mean ± SD. Two-tailed unpaired Student's $t$ test. **c** Western blotting images showing the expression of pSTAT3-Tyr705, SREBP2, and ENSA in three organoids. $n = 3$ independent experiments. **d, e** Results of the cell viability assay in three TNBC patient-derived organoid models treated with 0, 5, and 15 µM Stattic. **d** Cell viability assay and (**e**) Representative bright-field images of organoids after drug treatment in three organoids. Scale bars, 200 µm. $n = 3$. Data are presented as mean ± SD. Two-tailed unpaired Student's $t$ test. **f–g** Stattic treatment of MDA-MB-231 cells expressing control or ENSA shRNA. Briefly, we injected shCtrl or shENSA MDA-MB-231 cells into the mammary fat pad of female NOD/SCID mice ($n = 10$ each). When the tumor volume reached 50–100 mm$^3$, each group was randomly assigned to two treatment groups: vehicle and Stattic. All groups ($n = 5$ each) received treatment (vehicle or 10 mg/kg Stattic) three times per week after randomization. The gray arrows indicate the treatments. **f** The growth curve (left) and the inhibition rate of tumor volume (right). $n = 5$ mice per group. Data are presented as mean ± SD. Two-way ANOVA test for growth curve and two-tailed unpaired Student's $t$ test for inhibition rate. **g** The tumor weight (left) and the inhibition rate of tumor weight (right). $n = 5$ mice per group. Data are presented as mean ± SD. Two-tailed unpaired Student's $t$ test. **h** Scheme of the generation of the mini-PDX models for the in vivo pharmacological tests. **i** The relative viability of seven TNBC mini-PDX models with Stattic treatment, as normalized to vehicle treatment. $n = 3$ in low-ENSA group and $n = 4$ in high-ENSA group. Data are presented as mean ± SD. Two-tailed unpaired Student's $t$ test. **j** The relative luminance unit of each TNBC mini-PDX model treated with Stattic or vehicle. $n = 5$ or 6 independent capsules; Data are presented as mean ± SD. Two-tailed unpaired Student's $t$ test. Source data are provided as a Source Data file. n.s. not significant, Mini-PDX mini-patient-derived xenograft, RLU relative luminance unit.

TCACTACTAAA) and PPP2CA (siPPP2CA-1: CCGTGAACGCATCACCATT; siPPP2CA-2: GATACAAATTACTTGTTTA) were purchased from RiboBio. siRNA transfection was conducted with Lipofectamine RNAIMAX Transfection Reagent (Thermo Fisher Scientific) according to the manufacturer's instructions.

**Plasmid and cloning**. Human ENSA cDNA was purchased from GeneChem (Catalog number: NM_004436-GV492) and subcloned into the pCDH-CMV-MCS-EF1-puro plasmid (System Biosciences, Catalog number: CD510B-1). Human full-length STAT3 cDNA was purchased from Vigenebio (Catalog number: CH801341) and subcloned into the pCDH-CMV-MCS-EF1-GFP plasmid (modified from pCDH-CMV-MCS-EF1-puro). The SREBP2 promoter was amplified by using a pair of primers (forward: TGGTATTCCATCGTGTGGATGT; reverse: GAGTGAAGGGTTAACAGGCCA) in the BT549 cell line and cloned into the pGL3-basic vector (Promega). All transfections were performed using Lipofectamine 2000 transfection reagent (Thermo Fisher Scientific).

**Cell growth and colony formation assay**. For the cell growth assay, $2 \times 10^3$ (BT549, MDA-MB-231, and MCF7) and $4 \times 10^3$ (T47D, SKBR3, and BT474) cells were preseeded in 96-well plates in triplicate and incubated with 10% Cell Counting Kit-8 (CCK-8) solution (Vazyme, #A311-02) at 37 ℃ for 2 h, and then the absorbance was measured at 450 nm using a microplate reader. For the colony formation assay, $1 \times 10^3$ BT549 cells and $2 \times 10^3$ MDA-MB-231 cells were seeded into 6-well plates in triplicate, fixed after 12 days, and stained with 0.25% crystal violet staining solution. Colonies consisting of more than 50 cells were counted.

**Cell survival assay**. To assess the effect of chemicals on breast cancer viability, cells were grown in 96-well plates at $2 \times 10^3$ cells per well and exposed to different concentrations of the test chemicals. After 72 h, the cells were incubated with 10% CCK-8 (Vazyme, #A311-02) solution at 37 ℃ for 2 h. The cell survival percentages at different concentrations were calculated by dividing the optical density (OD) of chemical-containing wells by that of DMSO-contacting wells.

**Flow cytometry analysis**. For cell cycle analysis, a total of $1 \times 10^6$ cells were fixed with precooled 70% ethanol overnight and then processed using the Cell Cycle and Apoptosis Analysis Kit (Yeasen, #40301ES50) according to the manufacturer's instructions. For the cell apoptosis assay, $5 \times 10^5$ cells were collected and incubated with annexin V-fluorescin isothiocyanate (FITC) and propidium iodide (PI) staining solution from the Annexin V-FITC/PI Apoptosis Detection Kit (Yeasen, #40302ES50). The flow cytometry data were generated on a Beckman Cytomics FC 500 BD FACSCanto II and analyzed with FlowJo v10 software.

**Mouse models**. The animal protocols were approved by the Animal Welfare Committee of Shanghai Medical College at Fudan University (Protocol number: 20210510011). Female 6-week-old NOD.CB17-Prkdc scid/JSlac mice were used for the in vivo mouse xenograft models. Mice were exposed to 12 h light, 12 h darkness cycle at a temperature of 21 ± 3 ℃ and an average of 55% humidity. To evaluate the role of ENSA on tumor growth, $2 \times 10^6$ shCtrl or shENSA MDA-MB-231 cells were harvested and resuspended in a 100 µl volume (PBS: Matrigel=1:1) and then injected into the mammary fat pads of the mice ($n = 6$ each group). For the treatment groups, $2 \times 10^6$ shCtrl or shENSA MDA-MB-231 cells were injected into the mammary fat pads of the mice ($n = 5$ each group). When the tumor volumes reached 50–100 mm$^3$, vehicle or Stattic (10 mg/kg) was intraperitoneally administered three times a week for two weeks. The tumor volumes were calculated as follows: $V = L \times W^2 \times 1/2$, where $L$ is length (longest dimension) and $W$ is the

width (shortest dimension). After the endpoint, the mice were euthanized, and tumors were excised for analysis. Bioluminescence imaging was performed using the Multimodal Animal Rotation System (Bruker). Relative bioluminescence signal quantitation was performed by the respective imaging system software packages.

**Organoid**. Patient-derived organoids used in the current study were derived from post-surgery specimens of three female patients who underwent surgery at the Department of breast, Fudan University Shanghai Cancer Center. The organoids were cultured based on previously described methods[61,62]. The organoids were suspended in Basement Membrane Extract (BME) Type 2 (Trevigen, 3533-010-02) and cultured in breast cancer organoid medium (Advanced DMEM/F12 supplemented with R-spondin-1 [Peprotech], noggin [Peprotech], neuregulin [Peprotech], estradiol [Sigma], HEPES [Gibco], GlutaMAX [Gibco], nicotinamide [Sigma], N-acetylcysteine [Sigma], B-27 [Gibco], A83-01 [Tocris], primocin [InvivoGen], SB-202190 [Selleck], Y27632 [Selleck], FGF10 [Peprotech], FGF7 [Peprotech] and EGF [Peprotech]). After 3-5 passages, the organoids were added to each well of a 384-well plate, and different concentrations of Stattic (0, 5, and 15 µmol) were added to each well in duplicate and incubated for 5 days. Photos were taken on the last day to observe the changes in organoids under drug treatments. Acquisition of all clinical samples was approved by the Ethics Committee of FUSCC (Protocol number: 050432-4-1911D) and agreed to by each patient via signed informed consent.

**Mini-patient-derived xenograft (mini-PDX) model**. In vivo pharmacological tests were conducted using OncoVee mini-PDX assay (LIDE Biotech, Shanghai, China) according to the previous papers[24,25,60,63]. In brief, fresh surgical tumor specimens were acquired from seven female breast cancer patients (average age: 55 years) at FUSCC. Specimens were then washed with Hank's balanced salt solution (HBSS) to remove non-tumor tissues and necrotic tumor tissues. A fraction of tissue was retained for RNA extraction. The rest of the tissue was fragmented and digested with collagenase at 37 ℃ for 1–2 h. After centrifugation and removal of fibroblasts and blood cells with magnetic beads, tumor cells were collected and suspended to fill OncoVee capsules (LIDE Biotech, Shanghai, China). Each capsule contained 2000 cells and capsules derived from the same specimen were assigned to the baseline, vehicle control and Stattic treatment groups. Capsules were implanted subcutaneously into 5-week-old female nu/nu mice (3 capsules per mouse). Mice bearing capsules were treated with vehicle control or Stattic (10 mg/kg, intraperitoneal injection) for seven continuous days. Each treatment (vehicle control or Stattic) was performed in quintuplicate or sextuplicate capsules. Finally, the capsules were removed to measure cell viability in terms of relative luminance unit (RLU) using the CellTiter-Glo Luminescent Cell Viability Assay (Promega). Relative viability was calculated using the formula: Relative viability = (RLU of Stattic D7- RLU of bassline)/(RLU of vehicle D7- RLU of bassline) *100. Acquisition of all clinical samples was approved by the Ethics Committee of FUSCC (Protocol number: 050432-4-1911D) and agreed to by each patient via signed informed consent. The mini-PDX study protocol was approved by the Institutional Ethics Committee of Shanghai LIDE Biotech (Protocol number: LWIACUC002).

**RNA preparation and real-time quantitative reverse transcription (RT-qPCR)**. A RNeasy mini kit (Qiagen) was used for the purification of total RNA from breast cancer cells following the manufacturer's protocol. The extracted total RNA was subjected to cDNA synthesis using HiScript III RT SuperMix for qPCR (Vazyme). RT-qPCR was performed using ChamQ SYBR Color qPCR Master Mix (Vazyme) on a QuantStudio 6 Flex Real-Time PCR System (Applied Biosystems). The expression of genes was calculated using the $2^{-\Delta\Delta Ct}$ method, and the GAPDH was

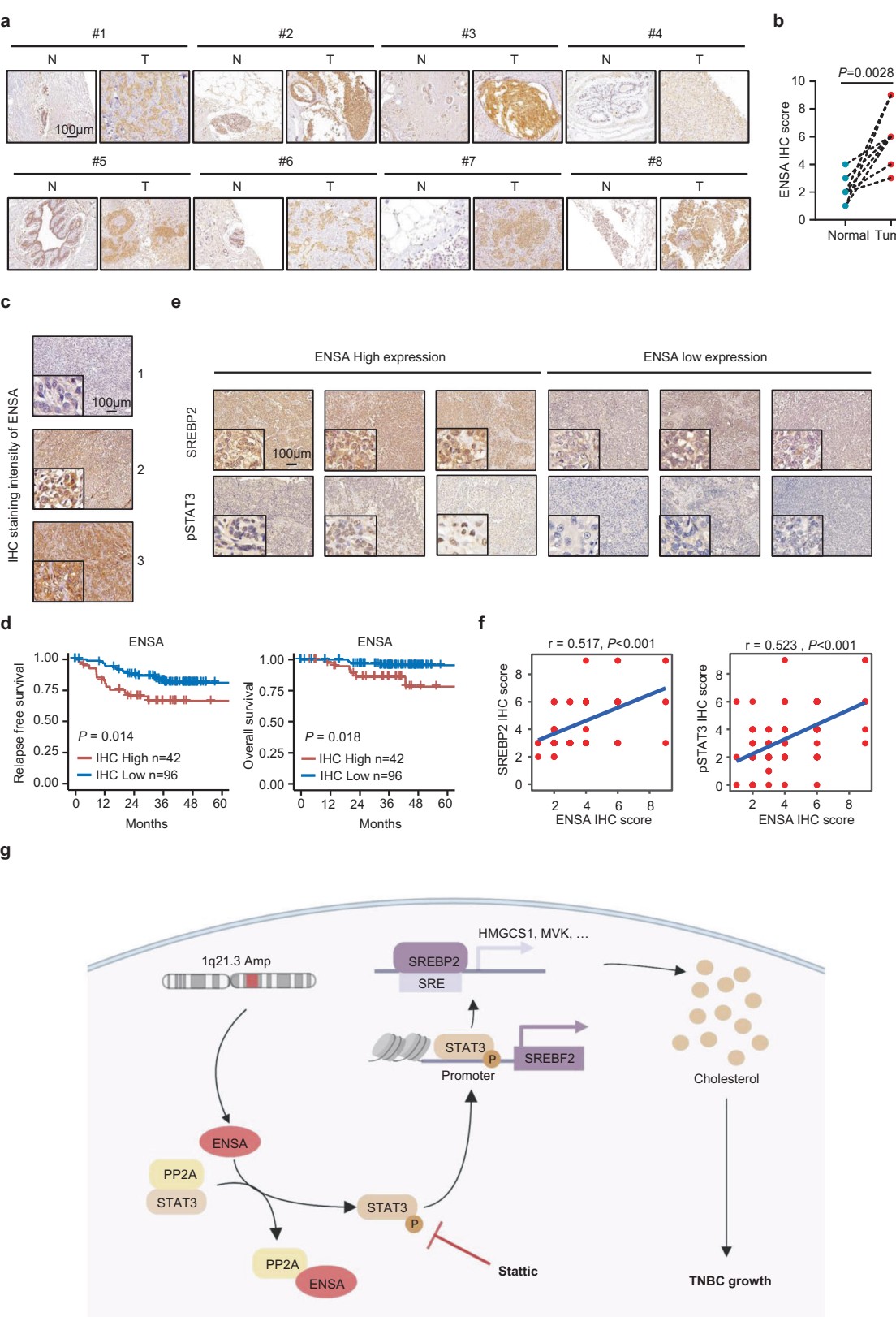

used for normalization. The sequences of the primers used for RT-qPCR are shown in Supplementary Table 5.

**RNA-seq and data analysis**. A total of 1 mg RNA samples were treated with VAHTS mRNA capture beads (Vazyme, China) to enrich polyA+ RNAs prior to constructing the RNA-seq libraries. The VAHTS mRNA-seq v2 library preparation

kit of the Illumina Xten system (Vazyme, Nanjing, China) was used to prepare the RNA-seq libraries according to the manufacturer's instructions. Briefly, polyA+ RNA samples (~100 ng) were fragmented and reverse transcribed into double strand cDNA. Then these cDNA fragments went through end repair, the addition of adenine tails, and ligation of the adaptors processes. The purified products were subjected to 12 cycles of PCR amplification to create the final cDNA libraries. These libraries were sequenced on 150 bp paired-end Illumina sequencing run.

**Fig. 7 Correlations among ENSA, SREBP2, pSTAT3-Tyr705 and survival in clinical samples. a, b** Representative IHC images (**a**) and IHC scores (**b**) of ENSA staining in 8 paired TNBC tissues and adjacent normal tissues. $n = 8$ paired samples. Two-tailed paired Student's $t$ test. **c** Immunohistochemical staining of ENSA in 138 TNBC specimens. Representative images are shown. Scale bars, 100 μm. **d** Kaplan–Meier analysis of the relapse-free survival and overall survival of 138 TNBC patients. A log-rank test was used to determine the statistical significance between the low-ENSA expression group ($n = 96$) and the high ENSA expression group ($n = 42$). **e** IHC staining of SREBP2 and pSTAT3-Tyr705 in 138 TNBC specimens. Representative images are shown. Scale bars, 100 μm. $n = 138$ samples. **f** Correlation analysis of ENSA with SREBP2 and pSTAT3-Tyr705 expression levels in 138 TNBC tissues. Correlation coefficients were calculated using the *Spearman* test. Two-tailed *P*-values were given. **g** Proposed working model. In TNBC, ENSA is amplified, highly expressed and inhibits the function of PP2A, resulting in STAT3 Tyr705 phosphorylation and activation. STAT3 activation induces SREBP2 transcription to upregulate cellular cholesterol biosynthesis and facilitate tumor progression. Inhibition of STAT3 signaling with Stattic might serve as an effective treatment strategy for 1q21.3-amplified and ENSA-highly expressed TNBC. Source data are provided as a Source Data file. IHC immunohistochemistry, Amp amplification, SRE sterol regulatory element.

Sequenced readings were aligned using HISAT2 with human genome GRCh38 as a reference genome. Gene expression levels were calculated from fragments per kilobase of transcript per million mapped reads (FPKM). Gene ontology (GO) analysis was performed using the Metascape tool, and the given input list contained genes that were expressed at a lower level than the shENSA group (fold change (FC) < 0.8). GSEA was performed using GSEA software (v3.0) and molecular signature database (v7.0).

**Cholesterol metabolism pathway analysis by LC-MS**. ENSA MDA-MB-231 cells ($n = 4$ each) ($1 \times 10^7$ shCtrl or shENSA) were collected for cholesterol analysis. Five hundred microliters of ethanol containing 5 μg of BHT were added to the cells. An internal standard cocktail (50 μL) comprising d6-lanosterol, d6-zymosterol, d7-desmosterol, d7-lathosterol, d7-d-dehydrocholesterol, and d6-cholesterol (Avanti Polar Lipids) was added to the samples. The samples were incubated at 1200 rpm for 15 min at 4 °C. At the end of incubation, 250 μL of Milli-Q water and 1 ml of n-hexane were added. The samples were mixed thoroughly by vortexing and then centrifuged at $15294 \times g$ for 5 min at 4 °C. The clear upper phase containing oxysterols and sterols in hexane was transferred to a new tube. The extraction was repeated once with another 1 ml of n-hexane. The pooled extract was dried in a SpeedVac under organic mode. Oxysterols and sterols were derivatized to obtain their picolinic acid esters prior to LC/MS analysis and quantitated by referencing the spiked internal standards as previously described[64]. The concentration of individual lipids (μmol/cell) was standardized to the z-score.

**Chemicals**. Cholesterol (#S4154) and Stattic (#S7024) were purchased from Selleck.

**Western blotting**. Total cellular protein was extracted using SDS lysis buffer [50 mM Tris (pH 8.1), 1 mM EDTA, 1% SDS, 1 mM fresh dithiothreitol, sodium fluoride, and leupeptin] and quantified using the BCA Protein Assay Kit (Solarbio). A total of 20 μg protein was separated by SDS-PAGE and then electrotransferred onto polyvinylidene difluoride membranes (Millipore). The membranes were incubated with the indicated primary antibody followed by an HRP-conjugated secondary antibody and then detected by enhanced chemiluminescence. For antibody use and details please see Supplementary Table 6.

**Filipin III staining**. A total of $3 \times 10^4$ cultured cells were preseeded in 24-well plates. Then, the cells were harvested, fixed with 4% paraformaldehyde and incubated with 0.05 mg/ml filipin III (Sigma, F4767) working solution for 2 h at room temperature. Then, the cells were sealed with SYTOX Deep Red stain (Invitrogen, P36990). This dye is excited by red light at 660 nm when bound to DNA and has an emission maximum at 682 nm; we detected these signals using a Cy5/deep red traditional filter. Filipin III staining of cells was visualized with a Leica DMI6000 B microscope at excitation wavelengths of 340–380 nm and emission wavelengths of 385–470 nm. The quantification of Filipin staining was performed by ImageJ (v1.8.0).

**Chromatin immunoprecipitation (ChIP)**. In brief, $1 \times 10^7$ cells were cross-linked with 1% formaldehyde and subjected to sonication in ChIP lysis buffer [50 mM HEPES (pH 7.5), 500 mM NaCl, 1 mM EDTA, 1% Triton X-100, and 0.1% Na-deoxycholate, supplemented with protease inhibitor cocktail]. Then, 4 μg anti-STAT3 or anti-mouse IgG antibodies with protein A/G magnetic beads (Invitrogen, #10015D) were added to each ChIP reaction for incubation. After three washes with lysis buffer, three washes with wash buffer (50 mM HEPES, 300 mM LiCl, 1 mM EDTA, 0.5% NP-40, and 0.7% Na-deoxycholate) and one wash with Tris-EDTA buffer (TE), each ChIP reaction was eluted and reverse cross-linked in elution buffer [50 mM Tris-HCl (pH 8.0), 10 mM EDTA, and 1% SDS] at 65 °C for 4 h. After RNase A and proteinase K digestion, DNA samples were isolated with phenol:chloroform:isoamyl alcohol (25:24:1) and analyzed by qRT-PCR. All results are displayed as fold change to 1% input. For antibody use and details please see Supplementary Table 6.

**Luciferase reporter assay**. A total of $5 \times 10^4$ cells were seeded in 24-well plates and transiently cotransfected with pGL3-SREBP2 promoter reporter plasmids and pRL-TK (Promega) using Lipofectamine 2000 transfection reagent. The firefly and Renilla luciferase activities were measured with a dual-luciferase reporter system (Promega) according to the manufacturer's instructions. The measurement was performed on a SpectraMax M5 Microplate Reader (Molecular Devices).

**Tissue specimens and immunohistochemistry (IHC)**. For IHC, 8 pairs of primary TNBC tissues and adjacent normal tissues and primary TNBC specimens from 138 female patients (average age: 54 years), who underwent surgery at FUSCC from 2010-2013 were obtained from the Department of Pathology, Fudan University Shanghai Cancer Center. The procedures for IHC were as follows: paraffin-embedded sections were deparaffinized at 60 °C for 4 h and then subjected to xylene and a graded series of alcohol. For antigen unmasking, the slides were heated with citrate or EDTA antigen retrieval solution. After cooling, the slides were blocked with blocking solution (2% goat serum, 2% bovine serum albumin, and 0.05% Tween 20 in PBS) for 10 min at room temperature (RT) and then incubated overnight with primary antibodies at 4 °C. The sections were covered with horseradish peroxidase (HRP)-conjugated secondary antibody (GeneTech) at RT for 30 min and then developed with 3,3′-diaminobenzidine substrate (Gene-Tech). The slides were counterstained with hematoxylin, dehydrated with a graded series of alcohols and then mounted with coverslips and mounting medium. The staining density was measured using a Leica CCD camera DFC420 connected to a Leica DM IRE2 microscope (Leica Microsystems Imaging Solutions Ltd.). Some staining images were scanned by PANNORAMIC MIDI (3DHISTECH Ltd.) and viewed with CaseViewer (v2.4). The IHC scores were calculated by multiplying staining intensity (0 = no, 1=weak, 2=moderate, 3=strong) with percentage of positive staining (0 = negative, $1 \leq 10\%$, $2 = 10–50\%$, $3 \geq 50\%$). Acquisition of all clinical samples was approved by the Ethics Committee of FUSCC (Protocol number: 050432-4-1911D) and agreed to by each patient via signed informed consent. For antibody use and details please see Supplementary Table 6.

**Statistics**. Statistical analysis was performed using SPSS (version 20.0), R software (version 4.0.3) and GraphPad (version 8.0.2). Two-way ANOVA was used to analyze the variance between two growth curves. One-way ANOVA and unpaired or paired Student's $t$ tests were used to compare data between two groups. Correlation coefficients were calculated using the Spearman test or Pearson test. The survival curves were generated by the Kaplan–Meier method and compared with the log-rank test. Multivariate Cox proportional hazard models provided calculated hazard ratios with 95% confidence intervals. Two-sided $P < 0.05$ was considered statistically significant.

**Reporting summary**. Further information on research design is available in the Nature Research Reporting Summary linked to this article.

## Data availability

RNA-seq data generated in this study are deposited in the Sequence Read Archive database under the accession number PRJNA713612. FUSCC TNBC sequence data were available in the NCBI Gene Expression Omnibus (OncoScan array; GSE118527) and Sequence Read Archive (whole-exome sequencing and RNA-seq; SRP157974). The expression data, CNA data, and clinical data of the TCGA cohort were downloaded from the cBioPortal website (https://www.cbioportal.org/). The expression data were then transformed according to the log2(RSEM + 1) method. The METABRIC expression data were downloaded from the cBioPortal website (https://www.cbioportal.org/). The expression data of the SMC cohort were available in the GEO database (GSE113184). Kaplan–Meier survival plots were generated online with the Kaplan–Meier plotter database (https://kmplot.com/analysis/), and hazard ratios with 95% confidence intervals and log-rank P values were calculated. The transcription factor binding site prediction was performed online with the JASPAR database (https://jaspar.genereg.net/). A public STAT3 ChIP-seq data were available in the GEO database (GSE152203). Source data are provided with this paper.

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

## Acknowledgements
The authors are grateful to Yi-Zhou Jiang and Ding Ma for their excellent data management. Cartoons in Figs. 6h, 7g were created with BioRender.com. This work was supported by grants from the National Natural Science Foundation of China (grants 81672600, 81722032, 82072916, and 91959207, received by K.-D.Y.), the 2018 Shanghai Youth Excellent Academic Leader (received by K.-D.Y.), and the Fudan ZHUOSHI Project (received by K.-D.Y.).

## Author contributions
K.-D.Y. and Y.-Y.C. designed the study; Y.-Y.-C. conducted and analyzed most of the experiments; J.-Y.G. and S.-Y.Z. conducted the remaining experiments; K.-D.Y. and Z.-M.S. provided crucial reagents and conduct management; Y.-Y.C. and K.-D.Y. prepared the manuscript. The authors read and approved the final manuscript.

## Competing interests
The authors declare no competing interests.
