## [Peer Review File · Nature Communications]

Copy number amplification of ENSA promotes the progression of triple-negative breast cancer via cholesterol biosynthesisREVIEWER COMMENTS

Reviewer #1 (Remarks to the Author); expert on cholesterol metabolism:

This is an interesting paper that identifies ESSA as a novel regulator of cholesterol biosynthesis by promoting the phosphorylation and activation of SREBP2, a master transcription factor, that controls the expression of genes involved in cholesterol biosynthesis and uptake. The authors found that copy numbers of ENSA are altered in triple-negative breast cancer leading to the activation of STAT3-regulation of SREBP2-mediated cholesterol metabolism.

There are some major questions that need to be answered to support the key conclusions of the authors:

1- The authors claimed that ENSA regulates cholesterol biosynthesis. However, this is not measured in the paper. This is a key experiment since the suppression of SREBP2 activity will influence cholesterol biosynthesis as well as other intermediates of the mevalonate pathway (e.g. farnesyl, geranylgeranyl, dolichol, etc) that are important in regulating cellular proliferation. Thus, the authors should measure sterol intermediates, de novo cholesterol synthesis in cells deficient in ENSA by ¹⁴C-acetate incorporation or LC/MS. Since cholesterol alone is not able to rescue colony formation in ENSA-depleted cells (Figure 3), this result suggests that other mevalonate intermediates play a key role in regulating cellular proliferation.

2- The effect of ENSA on cholesterol metabolism should be further studied. The authors should assess the expression of HMGCR, LSS and LDLR by Western Blot. Additional controls in some of the experiments are missing, including: a) the measurement of SREBP2 full protein in the study, the authors should rule out whether ENSA influences SREBP2 processing and nuclear translocation; b) suppression of cholesterol biosynthesis using statins in control and ENSA-deficient cells.

3- The analysis of clinical samples (Figure 6) is incomplete. The authors should measure the correlation of SREBP2-regulated genes (HMGCR, LDLR, etc) and ENSA expression.

Reviewer #2 (Remarks to the Author); expert on STAT3 and cancer therapy:

This manuscript by Yu et al. is an interesting article with a good mix of in vitro, in vivo, and human tissue studies to address prognostic copy number-associated genes and tease out a mechanism of action. The issue of copy number patterns being prognostic is becoming apparent in other tumor types, and can lead to the identification of novel genes contributing to the biology of these tumors. In this case, breast cancer. I am generally enthusiastic about this work. I do think the study could be strengthened somewhat by addressing the following:

1) As shown in Figure 1A, and described in the results, their genetic screen identified 41 genes in TNBC that have gene amplification and associated overexpression. It is unclear why the authors narrowed down their figure to show only 6 genes and even less clear of the rationale to focus their study on the 2 genes on 1q21.3 (and not other regions such as 1q43). How do the authors justify focusing on this narrow region and how do we know that they are not missing other potential important genes?

2) Results section, second paragraph: The authors state "...expression of ENSA, not GOLPH3L, was linked to poor relapse-free survival in TNBC and basal-like breast cancer (Fig. 1e, f)." However, GOLPH3L expression in basal-like breast cancer is not shown in Figure 1, and it is shown to be significant in TNBC.

3) All of the survival analysis for ENSA, etc. were performed using only univariate analysis (i.e. Kaplan-Meier analysis). To know whether ENSA amplification/expression is a true independent prognostic factor, the authors should apply multivariate analysis (Cox proportional hazard models) to account for

traditional prognostic factors such as axillary lymph node positivity, molecular subtype, histologic grade, lymphovascular invasion, tumor size, age, TNM stage, etc.

4) The authors show that overexpression of STAT3 is only a partial rescue of cell growth in ENSA depleted cells (Figure 4). This indicates that ENSA may act through both a STAT3-dependent and -independent mechanism. The authors have not addressed a STAT3-independent pathway for their ENSA-depleted phenotype.

5) The authors show efficacy of pharmacologic STAT3 inhibition in ENSA-depleted culture cells, but not in their in vivo fat pad injection mouse model. Because they have already performed in vivo experiments, it would be nice to know if STAT3 inhibition actually shows efficacy in their preclinical mouse model.

6) Line 192 incorrectly refers to Figure 1a, b and it should refer to Figure 6a, b.

7) The authors show some quantified immunohistochemical expression of ENSA/pSTAT3/SREBP2 in human breast tumors, but this is only limited to 6 patients and is therefore insufficient in numbers to claim these as a method of prognostication solely by IHC. It would be more convincing if a TMA or many more cases were provided for evidence. In addition, to increased numbers, it would be interesting to know whether these immunohistochemical patterns hold up in recurrent tumors as well as metastatic tumors.

Reviewer #3 (Remarks to the Author); expert on breast cancer:

In this manuscript, the authors have found a recurrent amplification in TNBC patients in which they have identified the gen ENSA that they associated with poor survival. The authors reveal that mechanistically, ENSA promotes the phosphorylation of STAT3 that upregulates SREBF2 expression, involved in cholesterol biosynthesis.

They conclude that ENSA-induced cholesterol biosynthesis favours TNBC tumor progression, which can be abrogated by treatment with STAT3 inhibitors.

The manuscript is well written, the results are adequately presented and the authors are able to portray the key proteins underlying the effects driven by the amplification of 1q21.3 region that is predominantly detected in TNBC patients. However, most conclusions rely on results done in vitro in only two cell models of TNBC and the manuscript would improve by further results performed in vivo. Moreover, key conclusions of the manuscript derive from IHC and IF images that do not have enough quality to be evaluated. Besides, the results using sttatic should be performed in the in vivo models of breast cancer cell xenografts.

Finally, the discussion lacks relevant points related to cholesterol and breast cancer. The data presented herein point to an altered cholesterol synthesis within the cells, how does the cholesterol from the diet would influence breast tumor progression?

General comments:

The criteria to select the two cell lines MDA-231 and BT549 is unclear. Do these cells show amplification of ENSA/ 1q21.3? The connection with cholesterol biosynthesis is only true in TNBC? Additional cell lines, TNBC but also luminal or HER2 should be included.

As 30% of patients with TNBC will present distant metastases during the course of breast cancer and metastasis in the cause of mortality in breast cancer. It is important to evaluate putative differences in metastasis in the shENSA xenograft models (MDA231 are highly metastatic). In addition several results are only shown in vitro. Validation in growing tumors in vitro is relevant.

Given the extensive collections of TNBC PDX models available in the field and their reliability as preclinical models the authors should validate their results using PDX models of TNBC patient with different levels of ENSA. They include some results with organoids but very poor. There is barely any information about the source of these organoids. How they were obtained ? Ethical information is

missing. Any characteristics of the tumors of origin (other than ENSA expression) will be relevant.

Specific points

Figure 1. In the figure 1g and 1h authors observed an increase of ENSA gene expression in TNBC. Other breast cancer subtypes as HR-HER2+ or HR+HER2+ present also an amplification of the same region identified in TNBC (Fig 1D). Authors should analyze whether ENSA expression associates with survival and /or alter the cholesterol biosynthesis in luminal or HER+ breast cancer and test the effect of ENSA depletion (shENSA) in luminal cell lines in terms of cell viability and cholesterol metabolism.

Figure 2: The authors show that ENSA expression rescues the phenotype induced by ENSA silencing in vitro, does it in vivo? Besides, do the tumors show more apoptosis in vivo (ie by Cleavage caspase 3 IHC on in paraffin tumor pieces). Why is the silencing of ENSA promoting apoptosis? Is this related to altered cholesterol biosynthesis? This is somehow addressed later but not in detail. This point should be clarified since it is relevant for the main conclusion of the work connecting breast cancer to cholesterol.

Figure 3. The images in Fig. 3E and F lack the quality, resolution and magnification required to draw conclusions. Does the addition of cholesterol rescue the proliferation and apoptosis phenotype seen upon ENSA silencing in the TNBC cell lines? Is it feasible to supplement the mice with cholesterol to rescue the decreased tumor growth seen in the in vivo assays upon ENSA silencing?

Figure 4. In the images shown in Fig. S4A is difficult to see the differences in pSTAT3 staining. The same is true for the IF displayed in Fig. 4E, no clear conclusions can be obtained from those images. Does the overexpression of STAT3 rescue the delayed tumor growth in vivo as well? Does the depletion of PP2A also lead to decrease SREBF2 levels?

Figure 5. Please add statistics to Figures 5A and 5B. shENSA cell lines that present decrease ENSA expression also show a small reduction in viability. Is it possible that static alter cell viability by other molecular mechanisms unrelated to ENSA/cholesterol biosynthesis? The efficiency of sttativ may be related with P-Stat3 levels and functionality. Does sttatic also reduce TNBC tumor growth in vivo? Were the levels of STAT3 and SREBF2 checked in the organoids shown in Fig. 5G?

Figure 6. The IHC panels shown in this figure are difficult to interpret; they pictures should have more resolution in order to see the proper stainings. Does SREBF2 levels also correlate with poor outcome in basal and TNBC cohorts such as those depicted in Fig. 1E and G? Do high ENSA and SREBF2 levels overlap in those cohorts? Do ENSA expression levels associate with cholesterol synthesis in TCGA patients and TNBC patients?

Minor points

Line 109: the adverb "However" does not fit at the beginning of the sentence since the results depicted are what would be expected.

Line 192: the results correspond to Fig. 6a, b instead of Fig. 1.

Aug 24, 2021

Dear Reviewers,

We are most grateful for your professional comments concerning our manuscript titled “**Copy number amplification of ENSA promotes the progression of triple-negative breast cancer via cholesterol biosynthesis**” (NCOMMS-21-07763). The comments were very helpful in revising and improving our paper and provided great guidance in our research. We are particularly pleased with the recognition of the novelty and rigor of our work noted by the Reviewers as follows: “This is an interesting paper that identify ESSA as a novel regulator of cholesterol biosynthesis” (from Reviewer 1). “This is an interesting article with a good mix of in vitro, in vivo, and human tissue studies to address prognostic copy number associated genes and tease out a mechanism of action” (from Reviewer 2). “The manuscript is well written; the results are adequately presented and the authors are able to portray the key proteins underlying the effects driven by the amplification of 1q21.3 region that is predominantly detected in TNBC patients.” (from Reviewer 3). We have studied the comments carefully and have comprehensively revised our manuscript to meet with approval. The revised portions have been marked in red in the file “**Manuscript Highlighted Changes**”. Our point-by-point responses to your comments are listed below.

Comment from Reviewer #1

Comment 1: *The authors claimed that ENSA regulates cholesterol biosynthesis. However, this is not measured in the paper. This is a key experiment since the suppression of SREBP2 activity will influences cholesterol biosynthesis as well other intermediates of the mevalonate pathway (e.g farnesyl, geranylgeranyl, dolichol, etc) that are important in regulating cellular proliferation. Thus, the authors should measure sterol intermediates, de novo cholesterol synthesis in cells deficient in ENSA by 14C-*

acetate-incorporation or LC/MS. Since cholesterol alone is not able to rescue colony formation in ENSA depleted cells (Figure 3), this result suggests that other mevalonate intermediates play a key role in regulating cellular proliferation.

Response: We greatly appreciate your valuable comment. As you mentioned, it is critical to validate the effect of ENSA on regulating de novo cholesterol biosynthesis in this article. According to your suggestion, we performed LC/MS analysis of cholesterol and several sterol intermediates from the cholesterol biosynthetic pathway (including squalene, lanosterol and metabolites in the downstream Bloch pathway and Kandutsch-Russell pathway) in control and ENSA-deficient TNBC breast cancer cells. Consistent with the downregulated mRNA and protein expression of genes involved in cholesterol biosynthesis, the concentration of cholesterol and most intermediates in the de novo cholesterol synthesis pathway was decreased when the expression of ENSA was suppressed. **We have made the appropriate revision in Fig. 3g, h and in the manuscript (page 8, lines 141–145).** These results supported our finding that the amplified gene ENSA played a vital role in de novo cholesterol biosynthesis.

Other questions from the reviewer were about the role of mevalonate intermediates in ENSA-regulated tumor growth. Regrettably, owing to limited metabolites in the cholesterol biosynthesis pathway testing panel (LipidALL Technologies, China), we were not able to directly quantify the level of several mevalonate intermediates, such as isopentenyl-diphosphate, farnesyl-diphosphate and geranylgeranyl-diphosphate. However, we can still infer that in addition to the critical function of altered cholesterol levels, other mevalonate intermediates and enzymes might partly account for the inhibited cellular proliferation in ENSA-depleted cells. Several mevalonate pathway metabolites and enzymes have been reported to be oncogenic in a variety of cancers, including breast cancer. Increased expression of multiple cholesterol biosynthesis proteins (EBP, HMGCS1, FDPS, FDFT1, DHCR7, and LSS) has been found in mammospheres, and these proteins correlated with the outcome of basal-like breast cancer patients (1). It

has been reported that FDPS plays an oncogenic role in PTEN-deficient prostate cancer through the GTPase/AKT axis (2). High-resolution CRISPR screens also identified several MVA pathway enzymes essential for the survival of cancer cells (3). In conclusion, mevalonate intermediates and enzymes might play a certain role in ENSA-regulated tumor growth, which explains the partial rescue function of cholesterol in ENSA-depleted cells. **We have added discussion to the manuscript (page 15, lines 277–299).**

Reference:

1. Ehmsen S, et al. Increased Cholesterol Biosynthesis Is a Key Characteristic of Breast Cancer Stem Cells Influencing Patient Outcome. *Cell Reports*. 2019;27(13):3927-38.e6.
2. Seshacharyulu P, et al. FDPS cooperates with PTEN loss to promote prostate cancer progression through modulation of small GTPases/AKT axis. *Oncogene*. 2019;38(26):5265-80.
3. Hart T, et al. High-Resolution CRISPR Screens Reveal Fitness Genes and Genotype-Specific Cancer Liabilities. *Cell*. 2015;163(6):1515-26.

Comment 2: The effect of ENSA on cholesterol metabolism should be also further study. The authors should assess the expression of HMGCR, LSS and LDLr by Western Blot. Additional controls in some of the experiments are missing this include: a) the measurement of SREBP2 full protein in the study, the authors should rule out whether ENSA influence SREBP2 processing and nuclear translocation; b) Suppression of cholesterol biosynthesis using statins in control and ENSA deficient cells.

Response: Thanks for your comment. We detected the expression of HMGCR, HMGCS1, LSS, FDFT1, and other enzymes involved in cholesterol biosynthesis in ENSA-depleted TNBC cells by western blotting. SREBP-2 is synthesized as a 125 kDa inactive precursor and sequentially cleaved into the NH₂-terminal form with nuclear translocation and transcription factor activity. To rule out whether ENSA influences

SREBP2 processing and nuclear translocation, we measured the protein expression of full and cleaved SREBP2. We found that the protein expression of both full and cleaved SREBP2 was decreased in ENSA-depleted cells, which was consistent with the altered mRNA level of SREBF2. **We have made the appropriate revision in Fig. 3f and in the manuscript (page 8, lines 135–137).**

Additionally, to illustrate the effect of ENSA on the cellular response to statins, we used statins in control and ENSA-depleted TNBC cells and measured cell viability. Compared with the control groups, the ENSA-depleted groups were more sensitive to the cholesterol biosynthesis inhibitor atorvastatin (**Fig. R1**). This phenotype might be explained by the inhibition of statin-induced feedback activation in the SREBP2 pathway upon ENSA silencing. It has been revealed that statins can induce SREBP2-dependent feedback gene activation, which underlies the resistance to statins by cancer cells (1). It has been uncovered that the inhibition of RAR-related orphan receptor gamma, which is important for SREBP2 recruitment and activation, could negate statin-induced, SREBP2-dependent feedback regulation and synergize with statins to kill TNBC cells (2). Similarly, the SREBP2 pathway was significantly suppressed upon ENSA depletion, which might account for the elevated sensitivity of the ENSA-depleted group to atorvastatin in the current study.

Figure R1: ENSA depletion induces the sensitivity of MDA-MB-231 and BT549 cells to atorvastatin.

Reference:

1. Mullen PJ, et al. The interplay between cell signalling and the mevalonate pathway in cancer. Nature Reviews Cancer. 2016;16(11):718-31.

2. Cai D, et al. ROR γ is a targetable master regulator of cholesterol biosynthesis in a cancer subtype.

Nature Communications. 2019;10(1):4621.

Comment 3: The analysis of clinical samples (Figure 6) is incomplete. The authors should measure the correlation of SREBP2 regulated genes (HMGCR, LDLR, etc) and ENSA expression.

Response: Thank you for your comment. We have added immunohistochemistry results of the SREBP2-regulated gene HMGCR and detected the correlation of HMGCR and ENSA expression. The results showed that ENSA expression and HMGCR expression showed a significant correlation ($r = 0.37, P < 0.001$). **We have made the appropriate revision in Fig. S8a-b and in the manuscript (page 12, line 227-230).**

Comment from Reviewer #2

Comment 1: As shown in Figure 1A, and described in the results, their genetic screen identified 41 genes in TNBC that have gene amplification and associated overexpression. It is unclear why the authors narrowed down their figure to show only 6 genes and even less clear of the rationale to focus their study on the 2 genes on 1q21.3 (and not other regions such as 1q43). How do the authors justify focusing on this narrow region and how do we know that they are not missing other potential important genes?

Response: Thank you for your thoughtful comments. We apologize for not making the statement of the screening criteria clear. According to the criteria listed in Fig. 1a, we found 41 copy number alteration (CNA)-affected oncogenic candidates in Fudan University Shanghai Cancer Center (FUSCC) TNBC. We listed the top 3 alteration peaks (1q21.3, 10p15.1 and 1q43) and six representative genes (two genes per region) without proper annotation before, which might cause some confusion. **We thus revised the schematic diagram in Fig 1a.** With regard to the rationale for focusing on two genes on 1q21.3, we considered the following two reasons: 1) among 41 candidates, two genes (*ENSA*, *GOLOH3L*) located in the chromosome 1q21.3 region had the highest rates of both amplification (56/302) and gain (174/302) events, which made us focus on this region. Genes with a high frequency of alteration are more likely to play vital roles in tumor progression and bring broader clinical potential. For example, HER-2-enriched breast cancer with a high frequency of HER2 amplification (80%) is a great clinical success for HER2-targeting therapy. 2) We also compared the survival difference of breast cancer patients with or without top 3 peak amplification. The results showed that neither 1q41.3 nor 10p15.1 amplification could bring a survival benefit for breast cancer patients, which was different from 1q21.3 amplification. For the above reasons, we focused on the amplified 1q21.3 region in the current study. **We have made the appropriate revision in Fig. S2a and in the manuscript (page 6, lines 92–93).**

Comment 2: Results section, second paragraph: The authors state "...expression of ENSA, not GOLPH3L, was linked to poor relapse-free survival in TNBC and basal-like breast cancer (Fig. 1e, f)." However, GOLPH3L expression in basal-like breast cancer is not shown in Figure 1, and it is shown to be significant in TNBC.

Response: We apologize for not showing the survival curve of *GOLPH3L* in basal-like breast cancer. **We have made the appropriate revision in Fig. 1f.** The results showed that high expression of *GOLPH3L* was correlated with better relapse-free survival of TNBC patients ($P = 0.018$). These results implied that *GOLPH3L* might be a cancer suppressor gene instead of an oncogene. We thus focused on *ENSA* rather than *GOLPH3L* in the current study.

Comment 3: All of the survival analysis for ENSA, etc were performed using only univariate analysis (ie Kaplan-Meier analysis). To know whether ENSA amplification/expression is a true independent prognostic factor, the authors should apply multivariate analysis (Cox proportional hazard models) to account for traditional prognostic factors such as axillary lymph node positivity, molecular subtype, histologic grade, lymphovascular invasion, tumor size, age, TNM stage, etc.

Response: Thank you for your suggestion. We performed multivariate analysis in 138 TNBC patients. The results showed that the high protein expression of *ENSA* was still correlated with poor relapse-free survival of TNBC patients after adjustment for tumor size, lymph node status and age. **We have made the appropriate revision in Supplementary Tables 2 and 3 and in the manuscript (page 12, lines 225–227).**

Comment 4: The authors show that overexpression of STAT3 is only a partial rescue of cell growth in ENSA depleted cells (Figure 4). This indicates that ENSA may act through both a STAT3-dependent and -

independent mechanism. The authors have not addressed a STAT3-independent pathway for their ENSA-depleted phenotype.

Response: Thank you for your thoughtful comment. In the current study, we illustrated that ENSA regulated TNBC progression partially via the pSTAT3-SREBF2 axis. Since STAT3 alone could rescue most of but not all the growth inhibition induced by ENSA depletion, other pathways might serve downstream of ENSA. As shown in **Figure 3a**, other important oncogenic pathways, such as the MYC pathway, were also enriched in ENSA-depleted TNBC cells. We thus inferred that ENSA might promote TNBC progression through multiple pathways, of which the STAT3-SREBF2 axis was the most important. The regulation of tumor progression in TNBC has been demonstrated to be complex, and one oncogene may promote TNBC progression via nonsingle downstream mechanisms. For example, androgen receptor promoted TNBC progression partially via AREG (1); oncogenic lncRNA TROJAN partially regulated TNBC progression via the ZMYND8 pathway (2). **We have added an additional discussion to the manuscript (page 17, lines 311–316).**

Reference:

1. Barton VN, et al. Multiple Molecular Subtypes of Triple-Negative Breast Cancer Critically Rely on Androgen Receptor and Respond to Enzalutamide In Vivo. *Mol Cancer Ther.* 2015;14(3):769.
2. Jin X, et al. The endogenous retrovirus-derived long noncoding RNA TROJAN promotes triple-negative breast cancer progression via ZMYND8 degradation. *Sci Adv.* 2019;5(3):eaat9820.

Comment 5: The authors show efficacy of pharmacologic STAT3 inhibition in ENSA-depleted culture cells, but not in their in vivo fat pad injection mouse model. Because they have already performed in vivo experiments, it would be nice to know if STAT3 inhibition actually shows efficacy in their preclinical mouse model.

Response: According to your suggestion, we performed in vivo experiments to investigate the effect of the STAT3 inhibitor Stattic in mammary fat pad injection xenograft models. We observed that the inhibition rates of both tumor volume and tumor weight were higher in the control group than in the ENSA-depleted group after Stattic treatment. We thus confirmed the efficacy of STAT3 inhibition in TNBC tumors with relatively high ENSA expression in preclinical models. **We have made the appropriate revision in Fig. 6f-g, Fig. S7d-e, and in the manuscript (page 11, lines 209–211; page 22, lines 410–411).**

Comment 6: Line 192 incorrectly refers to Figure 1a, b and it should refer to Figure 6a, b.

Response: Thank you for your comment. **We have made the appropriate revision in the manuscript (page 12, line 220).**

Comment 7: The authors show some quantified immunohistochemical expression of ENSA/pSTAT3/SREBP2 in human breast tumors, but this is only limited to 6 patients and is therefore insufficient in numbers to claim these as a method of prognostication solely by IHC. It would be more convincing if a TMA or many more cases were provided for evidence. In addition, to increased numbers, it would be interesting to know whether these immunohistochemical patterns hold up in recurrent tumors as well as metastatic tumors.

Response: Thank you for your comment. To investigate the correlation among ENSA, pSTAT3 and SREBF2 expression as well as prognostication in clinical samples, we actually performed IHC in 138 samples from primary sites of patients with TNBC (see Methods). The results were quantified by IHC scores and are shown with representative pictures. According to your suggestion, we have now provided more representative high-resolution pictures (**Fig. 7c, e and Fig. S8a**).

Since these 138 clinical samples were all collected from the primary sites of patients, we were unable to interrogate ENSA/pSTAT3/SREBP2 immunohistochemical patterns in recurrent or metastatic disease.

Instead, we assessed lung sections from fat pad xenograft models as a substitute to determine whether ENSA/pSTAT3/SREBP2 immunohistochemical patterns hold up in metastatic tumors. We observed decreased protein expression of pSTAT3 and SREBP2 in lung metastases of the ENSA-depleted group compared with the control group, which implied similar immunohistochemical patterns of ENSA/pSTAT3/SREBP2 in primary and metastatic tumors (**Fig. R2**).

Fig. R2: IHC staining of ENSA, pSTAT3 (Tyr507), and SREBP2 in lung metastases of MDA-MB-231 xenograft models.

Comment from Reviewer #3

The manuscript is well written, the results are adequately presented and the authors are able to portray the key proteins underlying the effects driven by the amplification of 1q21.3 region that is predominantly detected in TNBC patients. However, most conclusions rely on results done in vitro in only two cell models of TNBC and the manuscript would improve by further results performed in vivo. Moreover, key conclusions of the manuscript derive from IHC and IF images that do not have enough quality to be evaluated. Besides, the results using static should be performed in the in vivo models of breast cancer cell xenografts. Finally, the discussion lacks relevant points related to cholesterol and breast cancer. The data presented herein point to an altered cholesterol synthesis within the cells, how does the cholesterol from the diet would influence breast tumor progression?

Response: We appreciate the encouraging comments. We have added some additional experiments and made revisions according to comments from the editor and another reviewer to increase the credibility and scientific merit of the study. Importantly, **we have added additional discussion on cholesterol and breast cancer in the manuscript (page 15, lines 281–285)**. Point-by-point responses are listed as follows.

Comment 1: The criteria to select the two cell line MDA-231 and BT549 is unclear. Do these cells show amplification of ENSAX/ 1q21.3? The connection with cholesterol biosynthesis is only true in TNBC? Additional cell line, TNBC but also luminal or HER2 should be included.

Response: Thank you for your comment. We chose two TNBC cell lines, MDA-MB-231 and BT549, based on their high protein level of ENSA among multiple breast cancer cell lines. In addition to TNBC cells, the luminal cell lines MCF7 and T47D also showed relatively high ENSA expression, while the HER2 cell line showed relatively low ENSA expression. We thus further investigated the effects of ENSA depletion on non-TNBC cells and observed inhibited cell growth and an unaltered cholesterol biosynthesis pathway in these cells. (**See Response to Comment 4**).

Comment 2: As 30% of patients with TNBC will present distant metastases during the course of breast cancer and metastasis in the cause of mortality in breast cancer. It is important to evaluate putative differences in metastasis in the shENSA xenograft models (MDA231 are highly metastatic). In addition several results are only shown in vitro. Validation in growing tumors in vitro is relevant.

Response 2: Thank you for your thoughtful comment. We have now performed in vivo fat pad xenograft models with MDA-MB-231 cells to investigate the effect of ENSA knockdown on metastasis. We observed that the ENSA knockdown group had fewer lung metastases than the control group, suggesting that ENSA-

depleted TNBC cells had both decreased growth and metastasis ability in vivo. **We have made the appropriate revision in Fig. S5d and in the manuscript (page 10, lines 170–171).**

Comment 3: Given the extensive collections of TNBC PDX models available in the field and their reliability as preclinical models the authors should validate their results using PDX models of TNBC patient with different levels of ENSA. They include some results with organoids but very poor. There is barely any information about the source of these organoids. How they were obtained? Ethical information is missing. Any characteristics of the tumors of origin (other than ENSA expression) will be relevant.

Response 3: Patient-derived xenografts (PDXs) are translational preclinical models of cancer that recapture the characteristics of tumors in vivo. Unlike the efficient generation of PDXs with pancreatic cancer, liver cancer and brain tumors, the take rates of breast cancer PDXs have been 20% or less, which makes it difficult to yield successful xenografts within a limited time (1, 2). We really appreciate your suggestion on PDX models. The establishment of the PDX model is under way and will be applied in our future work. In the current study, we used organoids as substitute preclinical models for drug sensitivity testing. Breast cancer organoids can be generated efficiently, retain key characteristics of original tumors and serve as promising preclinical breast cancer models for cancer research (3, 4, 5). It has been illustrated that the in vitro drug responses of breast cancer organoids could match those of in vivo xenotransplantation and patients, suggesting the potential use of organoids as in vitro surrogates for breast cancer in vivo (3). The organoids used in the current study were derived from post-surgery specimens of patients who underwent surgery at the Department of Pathology, Fudan University Shanghai Cancer Center (FUSCC). Three stored TNBC organoids were resuscitated for drug sensitivity testing. The collection of all specimens for organoids was approved by the Ethics Committee of FUSCC, and informed consent documents were signed by all

patients. **Additional clinical characteristics of tumors of origins are listed in Table 1. We have made the appropriate revision in the manuscript (page 22, lines 417–419; page 29, line 570).**

Reference:

1. Yu J, et al. Establishing and characterizing patient-derived xenografts using pre-chemotherapy percutaneous biopsy and post-chemotherapy surgical samples from a prospective neoadjuvant breast cancer study. *Breast Cancer Research*. 2017;19(1):130.
2. Zhang X, et al. A renewable tissue resource of phenotypically stable, biologically and ethnically diverse, patient-derived human breast cancer xenograft models. *Cancer Research*. 2013;73(15):4885-97.
3. Sachs N, et al. A Living Biobank of Breast Cancer Organoids Captures Disease Heterogeneity. *Cell*. 2018;172(1-2):373-86.e10.
4. Arruabarrena-Aristorena A, et al. FOXA1 Mutations Reveal Distinct Chromatin Profiles and Influence Therapeutic Response in Breast Cancer. *Cancer Cell*. 2020;38(4):534-550.e9.
5. Sudhan DR, et al. Hyperactivation of TORC1 Drives Resistance to the Pan-HER Tyrosine Kinase Inhibitor Neratinib in HER2-Mutant Cancers [published correction appears in *Cancer Cell*. 2020 Feb 10;37(2):258-259]. *Cancer Cell*. 2020;37(2):183-199.e5.

Comment 4: Figure 1. In the figure 1g and 1h authors observed an increase of ENSA gene expression in TNBC. Other breast cancer subtypes as HR-HER2+ or HR+HER2+ present also an amplification of the same region identified in TNBC (Fig 1D). Authors should analyze whether ENSA expression associates with survival and /or alter the cholesterol biosynthesis in luminal or HER+ breast cancer and test the effect of ENSA depletion in luminal cell line in terms of cell viability and cholesterol metabolism.

Response: Thank you for your thoughtful comment. The detailed responses are listed as follows.

- 1) **Survival:** In the Kaplan-Meier plotter database, the survival curves of groups with high or low ENSA expression were not significantly different in the luminal A ($P=0.14$), luminal B ($P=0.19$), and HER2 ($P=0.18$) subgroups, which was different from the curves in the TNBC subgroup. Although luminal and HER2+ breast cancer also present amplification of the 1q21.3 region, the functions of ENSA in these subtypes may not be as important as those in the TNBC subtype considering the primary roles of estrogen and HER2 signaling in these two subtypes. **We have made the appropriate revision in Fig. S2b and in the manuscript (page 6, lines 100–102).**
- 2) **Cell growth and cholesterol biosynthesis:** To test the effect of ENSA depletion on cell growth and cholesterol biosynthesis in non-TNBC cells, we chose two luminal (MCF7 and T47D) and two HER2 (SKBR3 and BT474) cell lines for the in vitro assay. We observed slightly suppressed growth in ENSA-depleted luminal cells, less than the results shown in TNBC cells. Additionally, ENSA downregulation slightly impaired the cell growth of HER2 cells, in accord with relatively low ENSA expression in the HER2 cell line. We also performed western blotting to detect cholesterol pathway alterations following ENSA depletion. We found that the expression of pSREBP2, nSREBP2 and the downstream synthetase MVK was not altered by ENSA silencing in luminal and HER2 cell lines. These intriguing results indicate that ENSA might promote the growth of non-TNBC cells in an SREBP2- and cholesterol biosynthesis-independent manner. We are now working on another project to explore the underlying mechanisms of ENSA in non-TNBC cells. **We have made the appropriate revision in Fig. S3b, Fig. S4d, and the manuscript (page 7, lines 112–113; page 8, lines 138–141).**

Comment 5: Figure 2: The authors show that ENSA expression rescues the phenotype induced by ENSA silencing in vitro, does it in vivo? Besides, do the tumors show more apoptosis in vivo (ie by Cleavage caspase 3 IHC on in paraffin tumor pieces). Why is the silencing of ENSA promoting apoptosis? Is this

related to altered cholesterol biosynthesis? This is somehow addressed later but not in detail. This point should be clarified since it is relevant for the main conclusion of the work connecting breast cancer to cholesterol.

Response: Thank you for your thoughtful comment. The detailed responses are listed as follows.

- 1) **ENSA rescue in vivo.** We used mammary fat pad injection xenograft models to investigate the effect of ENSA expression on the phenotype induced by ENSA silencing in vivo. We observed that the inhibition of both tumor volume and tumor weight induced by ENSA depletion was rescued by ENSA expression, which confirmed the critical role of ENSA in promoting the growth of TNBC tumors in vivo. **We have made the appropriate revision in Fig. 4f-h, Fig. S5a-c and in the manuscript (page 9, lines 162–169).**
- 2) **Apoptosis in vivo.** To show tumor apoptosis in vivo, we performed IHC in tumors harvested from mammary fat pad-injected xenograft models. We observed higher apoptosis levels in the ENSA depletion group than in the control group and decreased apoptosis levels in the ENSA and STAT3 expression groups. **We have illustrated the results in Fig. 4h and have made the appropriate revision on manuscript page 9, lines 167–169.**
- 3) **Cholesterol biosynthesis and apoptosis.** In the current study, we found inhibited cholesterol biosynthesis and increased apoptotic levels in ENSA-depleted TNBC cells. To uncover the relationship between cholesterol biosynthesis and apoptosis upon ENSA silencing, we added experiments to investigate whether cholesterol could rescue the apoptosis induced by ENSA silencing. The results showed that the increased apoptosis level upon ENSA silencing could be partially rescued by the addition of cholesterol. **We have illustrated the results in Fig. S4e-f and have made the appropriate revision on manuscript page 9, lines 147–149.**

Comment 6: Figure 3. The images in Fig. 3E and F lack the quality, resolution and magnification required to draw conclusions. Does the addition of cholesterol rescue the proliferation and apoptosis phenotype seen upon ENSA silencing in the TNBC cell line? Is it feasible to supplement the mice with cholesterol to rescue the decreased tumor growth seen in the in vivo assays upon ENSA silencing?

Response: Thank you for your thoughtful comment. The detailed responses are listed as follows.

- 1) **Image quality.** For IHC images, we have now provided new images with higher quality (**Fig. 4h and Fig. 5g**). For IF images, we provided images with local magnification (**Fig. 3i**).
- 2) **Cholesterol rescue assay.** We have added experiments to determine whether the addition of cholesterol could rescue the phenotype induced by ENSA silencing. The results showed that suppressed cell growth and increased apoptosis levels upon ENSA silencing could be partially rescued by the addition of cholesterol, which was concordant with the results from the colony formation assay in **Fig. 3J**. **We have illustrated the results in Fig. S4e-f and have made the appropriate revision on manuscript page 9 and lines 147–149.**
- 3) **Cholesterol supplement in vivo.** There is no evidence for direct cholesterol supplementation in mouse models. Although some previous studies investigated the oncogenic role of cholesterol in tumor progression with high-fat diet (HFD)-fed mouse models, they barely discussed the role of de novo cholesterol biosynthesis within tumor cells (1, 2). For example, it has been shown that HFD-induced hypercholesterolemia could promote breast tumor metastasis in a tumor cell extrinsic manner dependent on 27-hydroxycholesterol (1). It is uncertain whether diet-derived cholesterol could be efficiently obtained by mammary tumor cells to rescue inhibited de novo cholesterol biosynthesis upon ENSA silencing in vivo. Thus, cholesterol supplementation experiments were performed in vitro for the current study.

Reference:

1. Baek AE, et al. The cholesterol metabolite 27 hydroxycholesterol facilitates breast cancer metastasis through its actions on immune cells. *Nature Communications*. 2017;8:864.
2. Rodrigues dos, et al. LDL-cholesterol signaling induces breast cancer proliferation and invasion. *Lipids in Health and Disease*. 2014;13(1):16.

Comment 7: Figure 4. In the images shown in Fig. S4A is difficult to see the differences in pSTAT3 staining.

The same is true for the IF displayed in Fig. 4E, no clear conclusions can be obtained from those images.

Does the overexpression of STAT3 rescue the delayed tumor growth in vivo as well? Does the depletion of PP2A also lead to decrease SREBF2 levels?

Response: Thank you for your thoughtful comment. The detailed responses are listed as follows.

- 1) **Image:** For IHC images, we have now provided new images with quantification, which showed a significant difference in pSTAT3 staining (**Fig. 4h**). For the IF images, we have provided more representative images with fluorescence quantitation as additional evidence (**Fig. 5f and Fig. S6c**).
- 2) **STAT3 rescue in vivo.** According to your suggestion, we used a mammary fat pad injection xenograft model to investigate the effect of STAT3 overexpression on the phenotype induced by ENSA silencing in vivo. We observed that the inhibition of both tumor volume and tumor weight induced by ENSA depletion was partially rescued by STAT3 expression, which was consistent with the in vitro results. **We have illustrated the results in Fig. 4f-h and Fig. S5a-c and have made the appropriate revision on manuscript page 9, lines 162–169.**
- 3) **PP2A.** PP2A is a phosphatase whose activity can be inhibited by ENSA. We performed a western blotting assay to detect SREBP2 levels upon PP2A depletion. The results showed that PP2A silencing could slightly increase SREBP2 levels and rescue SREBP2 levels in ENSA-depleted cells, which

implied the opposite effects of ENSA and PP2A on downstream STAT3-SREBP2 regulation. **We have illustrated the results in Fig. 5h and have made the appropriate revision on manuscript page 11, lines 195-197.**

Comment 8: Figure 5. Please add statistics to Figures 5A and 5B. shENSA cell line that present decrease ENSA expression also show a small reduction in viability. Is it possible that stactic alter cell viability by other molecular mechanisms unrelated to ENSA/cholesterol biosynthesis? The efficiency of sttatic may be related with P-Stat3 levels and functionality. Does sttatic also reduce TNBC tumor growth in vivo? Were the levels of STAT3 and SREBF2 checked in the organoids shown in Fig. 5G?

Response: Thank you for your thoughtful comment. The detailed responses are listed as follows.

- 1) **IC50.** We have added statistics to the IC50 curves and **illustrated the results in Fig. 6a and Fig. S7a.**
- 2) **Stattic in vitro.** It is possible that Stattic may inhibit cell viability in an ENSA/cholesterol biosynthesis-independent manner. As shown in **Fig. 4b**, ENSA-depleted cells showed a significant reduction in pSTAT3 levels but still expressed low levels of pSTAT3, which might lead to low sensitivity to Stattic.
- 3) **Stattic in vivo.** We used a mammary fat pad injection xenograft model to investigate the effect of the STAT3 inhibitor Stattic in vivo. We observed that the inhibition rates of both tumor volume and tumor weight were higher in the control group than in the shENSA group after Stattic treatment. We thus confirmed the efficacy of STAT3 inhibition in TNBC cells with high ENSA expression in our preclinical mouse models. **We have illustrated the results in Fig. 6f-g and Fig. S7d-e and made the appropriate revision on manuscript page 11, lines 209–211.**
- 4) **Organoids.** We performed western blotting to detect the levels of pSTAT3 and SREBP2 in the organoids shown in Fig. 5G. We observed higher expression of pSTAT3 and SREBP2 in organoids with higher ENSA expression. **We have illustrated the results in Fig. 6c.**

Comment 9: Figure 6. The IHC panels shown in this figure are difficult to interpret; they picture should have more resolution in order to see the proper stainings. Does SREBF2 levels also correlate with poor outcome in basal and TNBC cohorts such as those depicted in Fig. 1E and G? Do high ENSA and SREBF2 levels overlap in those cohorts? Do ENSA expression levels associate with cholesterol synthesis in TCGA patients and TNBC patients?

Response: Thank you for your thoughtful comment. The detailed responses are listed as follows.

- 1) **IHC images.** For the IHC images, we have now provided new images with higher quality. **We have illustrated the results in Fig. 7a, Fig. 7c, and Fig. 7e.**
- 2) **Survival.** We performed survival analysis of *SREBF2* expression on relapse-free survival of TNBC patients in the KM plotter database. The results showed that high *SREBF2* expression was correlated with worse relapse-free survival of TNBC patients. Since correlation analysis was not provided by the KM plotter database, we performed correlation analysis of *ENSA* and *SREBF2* in an external cohort, the SMC cohort (Korean breast cancer). It was a large cohort of Asian breast cancer enriched in younger premenopausal patients with accessible transcriptome and clinical data. The results showed that the expression of *ENSA* and *SREBF2* was positively correlated at the transcriptional level, which validated our findings in the current study. **We have illustrated the results in Fig. S8d-e and made the appropriate revision on manuscript page 12, lines 230–234.**
- 3) **External validation.** We performed ssGSEA in TCGA, METABRIC, and SMC cohorts to explore the correlation between *ENSA* expression and cholesterol biosynthesis activity in external populations. For patients with the TNBC subtype, *ENSA* expression was positively correlated with cholesterol biosynthesis in the TCGA ($P < 0.001$, $r = 0.243$), METABRIC ($P < 0.001$, $r = 0.308$), and SMC ($P = 0.047$, $r = 0.331$) cohorts, which further confirmed the clinical correlation between *ENSA* expression and

cholesterol biosynthesis in the current study. **We have illustrated the results in Fig. S4b and made the appropriate revision on manuscript page 8, lines 130–132.**

Comment 10: Line 109: the adverb “However” does not fit at the beginning of the sentence since the results depicted are what would be expected.

Response: Thank you for your suggestion. We have rewritten this sentence as suggested (**see manuscript, page 7, line 113**).

Comment 11: the results correspond to Fig. 6a, b instead of Fig. 1.

Response: Thank you for your comment. **We have made the appropriate revision in the manuscript (page 12, line 220).**

We would like to express our gratitude toward reviewers of our manuscript. We hope that the revised manuscript has addressed all of your questions and concerns. Thank you once again for your kind consideration.

REVIEWER COMMENTS

Reviewer #1 (Remarks to the Author):

The authors have addressed all my previous concerns

Reviewer #2 (Remarks to the Author):

The manuscript is improved substantially as the authors have addressed many of the comments of the reviewers. I am still not convinced that copy number of cholesterol synthesis related genes really drives the biology of this disease. But this manuscript will get that conversation going.

Reviewer #3 (Remarks to the Author):

The authors have addressed some of the concerns I raised but not the most important one. I appreciate the inclusion of additional analyses in luminal and HER2+ cell lines and survival analyses in these cohorts, and the improvement in the quality of the images. However, "in vivo" results are still poor and rely mainly on a single cell line, MDA-MB-231 cells, which are a poor representation of TNBC disease. This was the main reason to request experiments in PDX models. The intention was not that the authors generate PDX themselves; that is not feasible in the timeframe of the revision. However, there are many groups worldwide and consortia, that can provide TNBC PDX models, or collaborate in the studies.

Some additional concerns regarding the manuscript in its current format:

- Fig. 6F. Are the differences in tumor volume between shENSA+vehicle and shENSA+Stattic actually significant? From the graph shown, do not seem to be.

The authors' indeed claim that ENSA knockdown reduces sensitivity to Stattic treatment, and the significant differences should only be expected when shCtrl + vehicle is compared against all the other cohorts.

- Fig. 7F. The proposed working model is not self-explanatory, and the figure legend should be improved since Stattic is not even mentioned.

Minor points

- Fig. 6D: Stattic concentrations is written as μm instead of μM , same in the figure legend.

- Suppl. Fig. 4F: not properly labelled (shENSA instead of ENSA) and the figure legend should mention cholesterol addition.

- The sentence in Line 264 "resulting in the growth of the tumorsphere (28)" is odd.

Nov 12, 2021

Dear Reviewers,

We are most grateful for your professional comments concerning our manuscript titled “**Copy number amplification of ENSA promotes the progression of triple-negative breast cancer via cholesterol biosynthesis**” (NCOMMS-21-07763B). We have studied the comments carefully and have comprehensively revised our manuscript to meet with approval. The revised portions have been marked in red in the file “**Manuscript Highlighted Changes**”. Our point-by-point responses to your comments are listed below.

Comment from Reviewer #1

Comment 1: The authors have addressed all my previous concerns

Response: We greatly appreciate your recognition of our work.

Comment from Reviewer #2

Comment 1: The manuscript is improved substantially as the authors have addressed many of the comments of the reviewers. i am still not convinced that copy number of cholesterol synthesis related genes really drives the biology of this disease. But this manuscript will get that conversation going.

Response: Thank you for your comment. In the future, we’d like to focus on the relationship between metabolic dysregulation, especially cholesterol homeostasis dysregulation, with the development and progression of breast cancer.

Comment from Reviewer #3

Comment 1: In vivo results are still poor and rely mainly on a single cell line, MDA-MB-231 cells, which are a poor representation of TNBC disease. This was the main reason to request experiments in PDX models.

Response: Thank you for your thoughtful comment. To acquire more evidence from patient-derived in vivo models to support our findings, **we constructed seven mini-patient derived xenograft (mini-PDX) in vivo models** for pharmacological tests according to previous papers (Cancer Cell 2020;38:734-747; Nat Commun 2019;10:5492; Cancer Commun (Lond). 2018; 38: 48; Cancer Commun (Lond). 2018; 38: 60.), which were not only efficient but also feasible within the time of revision. As a novel alternative to the conventional PDX model, the mini-PDX model preserves the characteristics of patients' tumors and takes seven days for rapid drug sensitivity tests. It has been well established in several articles published in leading academic journals (Cancer Cell 2020;38:734-747. Nat Commun 2019;10:5492), which proved the effectiveness of these models. Several clinical trials have been carried out based on the mini-PDX model in view of its effectiveness and rapidity in providing guidance for prompt personalized treatment (metastatic TNBC: NCT04745975; pancreatic cancer: NCT04373928). Accordingly, we collected seven fresh post-surgery specimens from patients with TNBC and constructed mini-PDX models using the OncoVee mini-PDX assay (LIDE Biotech, Shanghai, China). The drug response to Stattic was detected after seven continuous days of administration, as normalized to vehicle treatment. We found that the relative viability of the high-ENSA group was lower than that of the low-ENSA group after Stattic treatment. Together, the results obtained from in vitro, in vivo and organoid models revealed that TNBCs with higher ENSA expression were more sensitive to STAT3 inhibitor Stattic. **We have illustrated the results in Figure 6h-6j, S7f and S7g and made the appropriate revision in manuscript page 12, line 215-222; page 23, line 442-458; page 30, line 601-604. The clinicopathological features of the patients and raw data for the mini-PDX assay were provided in Table S2.**

Reference:

1. Li C, Sun YD, Yu GY, Cui JR, Lou Z, Zhang H, et al. Integrated Omics of Metastatic Colorectal Cancer. *CANCER CELL*. 2020;38(5):734-47 e9.
2. Xu S, Zhan M, Jiang C, He M, Yang L, Shen H, et al. Genome-wide CRISPR screen identifies ELP5 as a determinant of gemcitabine sensitivity in gallbladder cancer. *Nat Commun*. 2019;10(1):5492.
3. Zhang F, Wang W, Long Y, Liu H, Cheng J, Guo L, et al. Characterization of drug responses of mini patient-derived xenografts in mice for predicting cancer patient clinical therapeutic response. *Cancer Commun (Lond)*. 2018;38(1):60.
4. Zhan M, Yang RM, Wang H, He M, Chen W, Xu SW, et al. Guided chemotherapy based on patient-derived mini-xenograft models improves survival of gallbladder carcinoma patients. *Cancer Commun (Lond)*. 2018;38(1):48.

Comments2: Fig. 6F. Are the differences in tumor volume between shENSA+vehicle and shENSA+Stattic actually significant? From the graph shown, do not seem to be. The authors' indeed claim that ENSA knockdown reduces sensitivity to Stattic treatment, and the significant differences should only be expected when shCtrl + vehicle is compared against all the other cohorts.

Response: Although not obvious in the figure, the difference in tumor volume between shENSA+vehicle and shENSA+Stattic groups was significant ($P < 0.05$). Similar to the in vivo drug response results, the shENSA group still showed a small reduction in viability when treated with Stattic in vivo. As shown in **Figure 4b**, ENSA-depleted cells showed a significant reduction in pSTAT3 levels but still expressed low levels of pSTAT3, which might lead to low sensitivity to Stattic. However, we indeed observed significantly impaired inhibition rate of tumor volume to Stattic when ENSA was depleted, as shown in **Figure 6f (right)**, which is in line with our expectation.

Comment 3: Fig. 7F. The proposed working model is not self-explanatory, and the figure legend should be improved since Static is not even mentioned.

Response: Thank you for your suggestion. **We have revised our working model in Figure 7g and made the appropriate revision in figure legend.**

Comment 4: Fig. 6D: Static concentrations is written as μm instead of μM , same in the figure legend.

Response: Thank you for your suggestion. **We have made the appropriate revision in Figure 6d and the figure legend.**

Comment 5: Suppl. Fig. 4F: not properly labelled (shENSA instead of ENSA) and the figure legend should mention cholesterol addition.

Response: Thank you for your suggestion. **We have made the appropriate revision in Figure S4f and the figure legend.**

Comment 6: The sentence in Line 264 “resulting in the growth of the tumorsphere (28)” is odd.

Response: Thank you for your suggestion. **We have made the appropriate revision in manuscript page 14, line 268-269.**

We would like to express our gratitude toward the reviewers of our manuscript. We hope that the revised manuscript has addressed all of your questions and concerns. Thank you once again for your kind consideration.

REVIEWER COMMENTS

Reviewer #3 (Remarks to the Author):

Which seems to be Table S2 does not contain clinicopathological features, looks like RLU readouts, maybe that comes from the conversion from excel to pdf, because another excel sheet 3 is missing. They claim that there is a significant difference in fig. 6F between shENSA+vehicle and +Stattic which is not the case.

I am not convinced about the value of mini-PDX as preclinical tools. Proper PDX models or additional cell lines are required

Jan. 7, 2022

Dear Reviewers,

We are most grateful for your professional comments concerning our manuscript titled “**Copy number amplification of ENSA promotes the progression of triple-negative breast cancer via cholesterol biosynthesis**” (NCOMMS-21-07763C). We have studied the comments carefully and have comprehensively revised our manuscript to meet with approval. The revised portions have been marked in red in the file “**Manuscript Highlighted Changes**”. Our point-by-point responses to your comments are listed below.

Comment from Reviewer #3

Comment 1: Table S2 does not contain clinicopathological features, looks like RLU readouts, maybe that comes from the conversion from excel to pdf, because another excel sheet 3 is missing.

Response: Thank you for your comment. We have provided information of Mini-PDX in Table S2 in Supplementary information file and provided the raw data of Mini-PDX assay in Source data file.

Comment 2: They claim that there is a significant difference in fig. 6F between shENSA+vehicle and +Stattic which is not the case.

Response: Thank you for your comment. To show the curves clearly, we split Y axis to highlight smaller data in Figure 6f. The difference between shENSA+vehicle and shENSA+Stattic groups was more obvious than that observed in untransformed image.

Figure R1. Related to Figure 6f.

Comment 3: I am not convinced about the value of mini-PDX as preclinical tools. Proper PDX models or additional cell lines are required.

Response: Thank you for your comment. We agree that PDX models are pretty important for preclinical validation. As alternative, Mini-PDX models show high consistency with PDX models, with specificity of 93% and sensitivity of 80%, and have been well established in several articles published in leading academic journals (Cancer Cell 2020;38:734-747. Nat Commun 2019; 10:5492). In the current study, we used seven TNBC Mini-PDX models, three TNBC organoid models, one cell line xenograft models to support in vitro and bioinformatics results. We acknowledge that standard PDX models can further validate our conclusions and have discussed the existing limitation of our study in the **manuscript page 18, line 343-347**.

We would like to express our gratitude toward the reviewers of our manuscript. We hope that the revised manuscript has addressed all of your questions and concerns. Thank you once again for your kind consideration.